# Low immunogenicity of malaria pre-erythrocytic stages can be overcome by vaccination

Katja Müller[1,2,†] (ID), Matthew P Gibbins[3,†,‡] (ID), Mark Roberts[3], Arturo Reyes-Sandoval[4,#] (ID), Adrian V S Hill[4] (ID), Simon J Draper[4] (ID), Kai Matuschewski[1,2] (ID), Olivier Silvie[5] (ID) & Julius Clemence R Hafalla[3,*] (ID)

## Abstract

Immunogenicity is considered one important criterion for progression of candidate vaccines to further clinical evaluation. We tested this assumption in an infection and vaccination model for malaria pre-erythrocytic stages. We engineered *Plasmodium berghei* parasites that harbour a well-characterised epitope for stimulation of CD8[+] T cells, either as an antigen in the sporozoite surface-expressed circumsporozoite protein or the parasitophorous vacuole membrane associated protein upregulated in sporozoites 4 (UIS4) expressed in exo-erythrocytic forms (EEFs). We show that the antigen origin results in profound differences in immunogenicity with a sporozoite antigen eliciting robust, superior antigen-specific CD8[+] T-cell responses, whilst an EEF antigen evokes poor responses. Despite their contrasting immunogenic properties, both sporozoite and EEF antigens gain access to antigen presentation pathways in hepatocytes, as recognition and targeting by vaccine-induced effector CD8[+] T cells results in high levels of protection when targeting either antigen. Our study is the first demonstration that poorly immunogenic EEF antigens do not preclude their susceptibility to antigen-specific CD8[+] T-cell killing, which has wide-ranging implications on antigen prioritisation for next-generation pre-erythrocytic malaria vaccines.

**Keywords** immunogenicity; malaria; *Plasmodium*; pre-erythrocytic; protective efficacy

**Subject Categories** Immunology; Microbiology, Virology & Host Pathogen Interaction

## Introduction

Malaria caused by the apicomplexan parasite *Plasmodium* is responsible for more than 229 million clinical cases and over 409,000 deaths annually worldwide, with more than 97% of cases attributable to *Plasmodium falciparum* (WHO, 2020). Whilst current malaria control strategies have led to marked reduction in incidence rate, cases and mortality for the past 16 years, a highly efficacious vaccine is likely essential to approach the ambitious World Health Organisation's (WHO) vision of "a world free of malaria". Targeting the malaria pre-erythrocytic stages, an obligatory and clinically silent phase of the parasite's life cycle, is considered an ideal and attractive strategy for vaccination; inhibiting parasite infection of, and development in hepatocytes results in preclusion of both disease-causing blood stages and transmissible sexual stages. Yet, despite intensive research for over 35 years, a highly efficacious pre-erythrocytic stage vaccine remains elusive (Draper *et al*, 2018). An in-depth characterisation of how the complex biology of pre-erythrocytic stages influences the generation and protective efficacy of immune responses is warranted to inform the design of future malaria vaccines.

CD8[+] T cells are crucial mediators of protective immunity to malaria pre-erythrocytic stages (Doolan & Hoffman, 2000). Whilst often considered as a single phase of the parasite's life cycle, the malaria pre-erythrocytic stage is comprised of two different parasite forms: (i) sporozoites, which are motile extracellular parasites that are delivered by infected mosquitoes to the mammalian host, and (ii) exo-erythrocytic forms (EEF; also known as liver stages), which are intracellular parasites resulting from the differentiation and growth of sporozoites inside a parasitophorous vacuole (PV) within hepatocytes (Hafalla *et al*, 2011). How these two spatially different parasite forms and the ensuing temporal expression of parasite-derived antigens impact the magnitudes, kinetics and phenotypes of CD8[+] T-cell responses elicited following infection is poorly understood. Furthermore, the complexity within the pre-erythrocytic stages has fuelled a long-standing debate focussed on the contributions of distinct sporozoite and EEF antigens in parasite-induced responses, and whether sporozoite or EEF proteins are better targets of vaccines.

Our current understanding of CD8[+] T-cell responses to malaria pre-erythrocytic stages has been largely based on measuring

1    Parasitology Unit, Max Planck Institute for Infection Biology, Berlin, Germany
2    Department of Molecular Parasitology, Institute of Biology, Humboldt University, Berlin, Germany
3    Department of Infection Biology, Faculty of Infectious and Tropical Diseases, London School of Hygiene and Tropical Medicine, London, UK
4    Jenner Institute, University of Oxford, Oxford, UK
5    Sorbonne Université, INSERM, CNRS, Centre d'Immunologie et des Maladies Infectieuses, CIMI-Paris, Paris, France
     *Corresponding author. Tel: +44 020 7958 8129; E-mail: julius.hafalla@lshtm.ac.uk
     †These authors contributed equally to this work
     ‡Present address: Wellcome Centre for Integrative Parasitology, Institute of Infection, Immunity and Inflammation, University of Glasgow, Glasgow, UK
     #Present address: Instituto Politécnico Nacional, IPN. Av. Luis Enrique Erro s/n, Unidad Adolfo López Mateos, Mexico City, Mexico

responses to the H-2-K$^d$-restricted epitopes of rodent *P. yoelii* (*Py*) (Weiss *et al*, 1990) and *P. berghei* (*Pb*) (Romero *et al*, 1989) circumsporozoite proteins (CSP), the major surface antigen of sporozoites. Many of these fundamental studies have focussed on using infections with irradiated sporozoites, the benchmark vaccine model for malaria. Infection with *Py* sporozoites elicits an expected T-cell response typified by early activation and induction of effector CSP-specific CD8$^+$ T cells followed by contraction and establishment of quantifiable memory populations (Sano *et al*, 2001). CSP-specific CD8$^+$ T cells are primed by dendritic cells that cross-present sporozoite antigens via the endosome-to-cytosol pathway (Cockburn *et al*, 2011). Yet, CSP is a unique antigen because it is expressed in both sporozoites and EEFs (Hollingdale *et al*, 1983). Whilst the expression of CSP mRNA ceases after sporozoite invasion, the protein on the parasite surface is stable and endures in EEFs during development in hepatocytes (Silvie *et al*, 2014). *In vitro* data indicate that primary hepatocytes process and present *Pb*CSP-derived peptides to CD8$^+$ T cells in a proteasome-dependent manner, involving export of antigen to the cytosol (Cockburn *et al*, 2011). Taken together, these data imply that sporozoite antigens induce quantifiable CD8$^+$ T-cell responses after infection. Antigens that have similar expression to the CSP, persisting to EEFs and with epitope determinants presented on hepatocytes, are excellent targets of CD8$^+$ T cell-based vaccines.

The paucity of EEF only-specific epitopes has hindered not only our ability to understand the immune responses that are evoked whilst the parasite is in the liver, but also their utility as targets of vaccination. Accordingly, the contribution of EEF-infected hepatocytes in the *in vivo* induction of CD8$^+$ T-cell responses is poorly understood. The liver is an organ where the primary activation of CD8$^+$ T cells is generally biased towards the induction of tolerance (Thomson & Knolle, 2010; Bertolino & Bowen, 2015). Yet, studies in other model systems have demonstrated antigen-specific primary activation within the liver (Bertolino *et al*, 2001). Another confounding issue with EEFs is their development in PVs with constrained access to the hepatocyte's cytosol (Hafalla *et al*, 2011). Nonetheless, if CD8$^+$ T cells specific for EEF antigens are primed, do they expand and contract with distinct kinetics? Moreover, are EEF-specific epitopes efficiently generated for recognition and targeting by vaccine-induced CD8$^+$ T cells? Answers to these questions will be key for antigen selection and design of future malaria vaccines.

In this study, we compared the initiation and development of CD8$^+$ T-cell responses—elicited following parasite infection—to CSP, a sporozoite antigen, and to upregulated in infective sporozoites gene 4 (UIS4), an EEF-specific vacuolar protein (Mueller *et al*, 2005). UIS4, a member of the early transcribed membrane protein (ETRAMP) family, is abundantly expressed in EEFs and associates with the PVM (Mueller *et al*, 2005). Whilst UIS4 mRNA expression is present in sporozoites, translation is repressed until when EEFs develop (Silvie *et al*, 2014). To control for epitope specificity, we generated *Pb* transgenic parasites that incorporate the MHC class I H-2-K$^b$ epitope SIINFEKL, from ovalbumin, in either the CSP or UIS4 protein. The resulting transgenic parasites develop normally as wild-type (WT) *Pb* in the mosquito vector and mammalian host. However, SIINFEKL would be expressed at the same time and space as its respective *Plasmodium* protein, enabling the CD8$^+$ T-cell response against these proteins to be tracked in an epitope-specific physiological manner. In line with previous studies (Cockburn *et al*,

2011; Montagna *et al*, 2014), to augment low numbers of CD8$^+$ T cells in the naïve response, cells from OT-I mice, which express SIINFEKL-specific T-cell receptors (TCRs) on their CD8$^+$ T cells, were initially adoptively transferred to mice prior to them receiving sporozoite immunisations. Furthermore, we evaluated the capacity of vaccine-induced CD8$^+$ T cells to target these parasites in a mouse challenge model. Our data show disparate immunogenic properties between a sporozoite and an EEF vacuolar membrane antigen but equivalent susceptibility to vaccine-induced CD8$^+$ T cells.

# Results

## Transgenic CSP$^{SIINFKEL}$ and UIS4$^{SIINFEKL}$ parasites display normal sporozoite motility and liver invasion

We generated, by double homologous recombination, transgenic *Pb* parasites expressing the immunodominant H-2-K$^b$-restricted CD8$^+$ T-cell epitope of ovalbumin (SIINFEKL) in the context of the sporozoite surface antigen CSP or the EEF vacuolar membrane antigen UIS4 (Figs 1A and EV1A and B). Constructs included the *TgDHFR/TS*-positive selection cassette and incorporated SIINFEKL in the context of the gene open reading frame. For CSP$^{SIINFEKL}$, SIINFEKL replaced SYIPSAEKI, the immunodominant H-2-K$^d$-restricted CD8$^+$ T-cell epitope of CSP, which allowed for recognition in H-2-K$^b$-carrying C57BL/6 mice. For UIS4$^{SIINFEKL}$, the SIINFEKL epitope was added to the immediate C terminus of the UIS4 protein. Appending the C terminus was chosen because it had been shown in *Toxoplasma gondii* that the potency of the immunodominant epitope of GRA6 was associated with its C-terminal location, which may have enhanced the presentation by parasite-infected cells (Feliu *et al*, 2013). Whilst undefined for UIS4 itself, it has been shown for several other ETRAMPs that the C terminus faces the host cell cytoplasm (Spielmann *et al*, 2003), which is likely to facilitate exposure to the MHC I complex.

The resulting parasites showed a phenotype comparable to WT parasites, with comparable mosquito infectivity and number of salivary gland sporozoites (Fig EV1C and D), functional sporozoite motility (Fig 1B) and normal invasive capacity and development inside hepatocytes (Figs 1C and EV1E). Thus, the introduced mutations to generate CSP$^{SIINFEKL}$ and UIS4$^{SIINFEKL}$ parasites did not interfere with the completion of the life cycle, in either mosquito vector or mouse. All C57BL/6 mice that received 800 sporozoites of either CSP$^{SIINFKEL}$ or UIS4$^{SIINFEKL}$ intravenously developed a detectable (patent) blood stage infection by day 4, comparable to infection with WT sporozoites (Fig EV1F).

## Peripheral blood CD8$^+$ T-cell responses are superior if elicited by a sporozoite surface protein in contrast to a vacuolar membrane protein in the infected liver

We first wanted to determine whether the generated transgenic parasites allow antigen-specific responses to be tracked using SIINFEKL as a surrogate CD8$^+$ T-cell epitope for sporozoite surface and EEF vacuolar membrane antigens. To this end, we assessed the kinetics of the CD8$^+$ T-cell response following intravenous immunisation with CSP$^{SIINFEKL}$ or UIS4$^{SIINFEKL}$ sporozoites. To augment the CD8$^+$ T-cell response, mice were adoptively transferred with $2 \times 10^6$

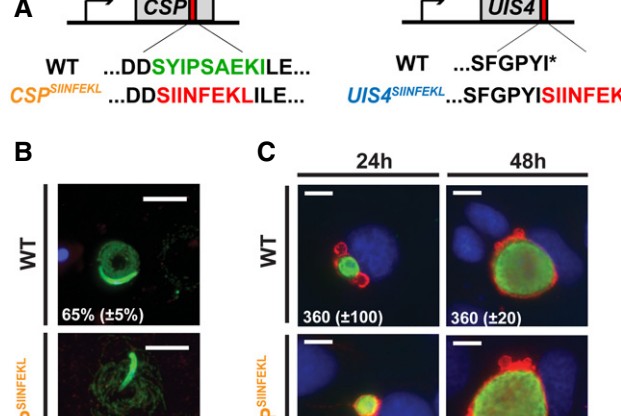

**A** Schematic overview of transgenic parasite lines:

WT ...DD**SYIPSAEK**ILE...
*CSP*<sup>SIINFEKL</sup> ...DD**SIINFEKL**ILE...

WT ...SFGPYI*
*UIS4*<sup>SIINFEKL</sup> ...SFGPYI**SIINFEKL***

**B**

WT — 65% (±5%)
CSP<sup>SIINFEKL</sup> — 71% (±16%)
UIS4<sup>SIINFEKL</sup> — 55% (±12%)

UIS4 CSP Hoechst

**C**

24h / 48h

WT — 360 (±100) / 360 (±20)
CSP<sup>SIINFEKL</sup> — 220 (±10) / 230 (±20)
UIS4<sup>SIINFEKL</sup> — 370 (±40) / 430 (±40)

UIS4 HSP70 Hoechst

Figure 1. **Generation and characterisation of recombinant CSP**<sup>SIINFEKL</sup> **and UIS4**<sup>SIINFEKL</sup> *P. berghei* **parasites.**

*Plasmodium berghei* parasites expressing the CD8$^+$ T-cell epitope of ovalbumin, SIINFEKL, in the context of CSP or UIS4 were generated using double homologous recombination.

A   Schematic overview of transgenic parasite lines. To generate CSP<sup>SIINFEKL</sup>, SIINFEKL replaced amino acids SYIPSAEK in CSP. To generate UIS4<sup>SIINFEKL</sup>, SIINFEKL was adjoined to the carboxyl-terminus of the UIS4 protein.
B   Sporozoite immunofluorescent antibody staining of WT, CSP<sup>SIINFEKL</sup> or UIS4<sup>SIINFEKL</sup> sporozoites after gliding on BSA-coated glass slides. Shown are microscopic images of the respective sporozoites that were stained with anti-CSP (green), anti-UIS4 (red) and nuclear stain Hoechst (blue). Scale bars, 10 μm. The numbers show mean percentage (± SD) of sporozoites with trails assessed from ≥ 220 sporozoites.
C   Fluorescent-microscopic images of EEF-infected Huh7 hepatoma cells. 24 and 48 h after infection with WT, CSP<sup>SIINFEKL</sup> or UIS4<sup>SIINFEKL</sup> sporozoites, the cells were fixed and stained with anti-UIS4 (red), anti-HSP70 (green) and the nuclear stain Hoechst (blue). Scale bars, 10 μm. The numbers show mean numbers (± SD) of intracellular parasites counted per well of 8-well Labtek slides.

Data information: The data shown are representative from one of two independent experiments.

OT-I cells expressing a SIINFEKL-specific TCR (Cockburn *et al*, 2011), prior to receiving 10,000 γ-radiation attenuated WT, CSP<sup>SIINFEKL</sup> or UIS4<sup>SIINFEKL</sup> sporozoites. Prior work showed that γ-radiation attenuation of *P. berghei* sporozoites does not impact host cell invasion and UIS4 expression (Mac-Daniel *et al*, 2014).

Peripheral blood was taken at days 4, 7, 14, 21, 42 and 88 after immunisation and CD8$^+$ T-cell responses were analysed after staining with H-2-K$^b$-SIINFEKL pentamers and for CD11a, a marker for antigen-experienced T cells (Rai *et al*, 2009; Schmidt *et al*, 2010) (Fig 2A and B). Additional gating strategies for the characterisation of K$^b$-SIINFEKL$^+$ CD8$^+$ T cells are shown in Appendix Fig S1. A substantial proportion of K$^b$-SIINFEKL$^+$ CD11a$^{hi}$ CD8$^+$ T cells were observed in mice immunised with CSP<sup>SIINFEKL</sup>; the response was highest on day 4, reaching 5% of all antigen-experienced CD8$^+$

T cells, and declined steadily until day 21, when the response stabilised and remained unchanged for several weeks (Fig 2C). In marked contrast, UIS4<sup>SIINFEKL</sup> immunisation induced a poor CD8$^+$ T-cell response; the proportion of K$^b$-SIINFEKL$^+$ CD11a$^{hi}$ CD8$^+$ T cells was only higher than the control groups at day 4 after immunisation, and the response remained within background levels for the duration of the experiment. Control groups included mice receiving OT-I cells only or in addition to γ-radiation attenuated WT sporozoites, which lack SIINFEKL sequences.

The poor CD8$^+$ T-cell response induced by UIS4<sup>SIINFEKL</sup> sporozoites, as compared to CSP<sup>SIINFEKL</sup>, led us to characterise the early events in the proliferation and differentiation of these cells. Mice were adoptively transferred with CFSE-labelled OT-I cells and immunised with γ-radiation attenuated WT, CSP<sup>SIINFEKL</sup> or UIS4<sup>SIINFEKL</sup> sporozoites. As shown by gating on CD8$^+$ T cells (Fig 2D and E), after 5 days, immunisation with CSP<sup>SIINFEKL</sup> sporozoites led to greater expansion of K$^b$-SIINFEKL$^+$ CD8$^+$ T cells, as compared to that observed with UIS4<sup>SIINFEKL</sup> sporozoites (Fig EV2A), in good agreement with the peripheral blood data described above (Fig 2C). Consistent with the activation of these cells, the proliferation of antigen-specific CD8$^+$ T cells by both parasites was associated with the development of effector and effector memory phenotypes as evidenced by upregulation of CD11a and CD49d, and downregulation of CD62L, respectively (Figs 2F–H and EV2B–D).

Taken together, these findings establish that immunisations with γ-radiation attenuated CSP<sup>SIINFEKL</sup> and UIS4<sup>SIINFEKL</sup> sporozoites permit antigen-specific responses to be tracked longitudinally in the peripheral blood. Importantly, we demonstrate that a sporozoite surface protein evokes a CD8$^+$ T-cell response of superior magnitude than an EEF vacuolar membrane protein following immunisation with malaria sporozoites.

## Higher magnitude of splenic and intrahepatic CD8$^+$ T-cell responses to a sporozoite antigen

Previous research has shown that CD8$^+$ T cells are primed primarily in the spleen following intravenous immunisation with malaria sporozoites (Lau *et al*, 2014) and that liver lymphocytes form a front-line defence against developing EEFs in hepatocytes (Guebre-Xabier *et al*, 1999; Fernandez-Ruiz *et al*, 2016). Thus, we further analysed the development of CD8$^+$ T-cell responses in the spleens and livers of mice adoptively transferred with OT-I cells and intravenously immunised with γ-radiation attenuated WT, CSP<sup>SIINFEKL</sup> or UIS4<sup>SIINFEKL</sup> sporozoites. Consistent with our aforementioned results, surface staining of splenic and liver lymphocytes showed higher proportion and absolute numbers of K$^b$-SIINFEKL$^+$ CD11a$^{hi}$ CD8$^+$ T cells at day 14 and day 42 following immunisation with CSP<sup>SIINFEKL</sup> compared with UIS4<sup>SIINFEKL</sup> sporozoites (Fig 3A–D). In addition, the proportions of CD11a$^{hi}$, CD62L$^{lo}$, CD49d$^{hi}$ and CD44$^{hi}$ expressing splenic and liver CD8$^+$ T cells, elicited by both CSP<sup>SIINFEKL</sup> and UIS4<sup>SIINFEKL</sup> sporozoites, were comparable, indicative of effector and effector memory cell phenotypes (Fig EV3). Although low, the numbers of antigen-specific CD8$^+$ T cells induced by UIS4<sup>SIINFEKL</sup> sporozoites were within the detection limits of the assay.

To assess for effector functions, splenic and liver lymphocytes were stimulated *ex vivo* with the SIINFEKL peptide. Generally,

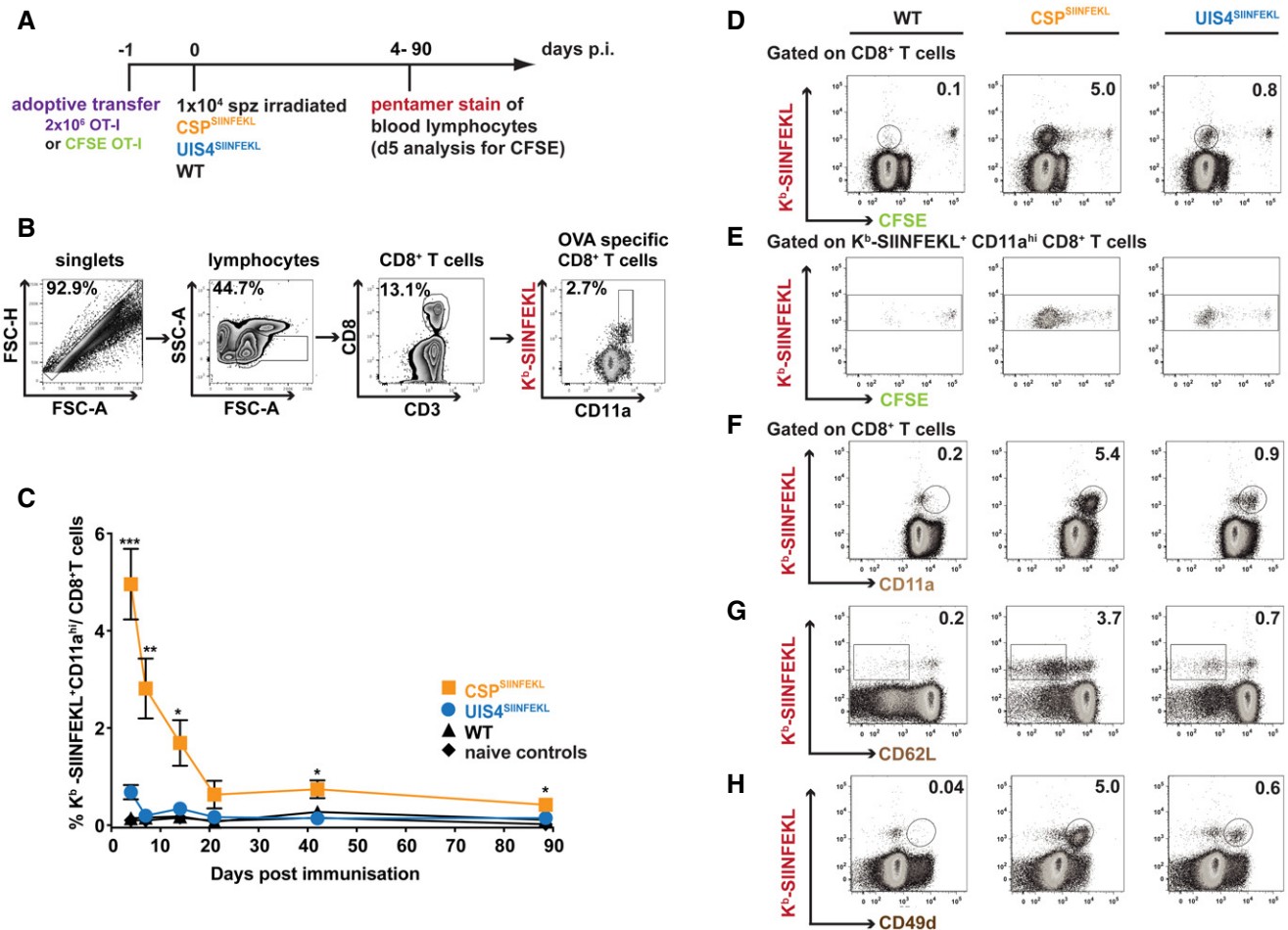

**Figure 2. Sporozoite antigen exposure elicits superior peripheral blood CD8+ T-cell responses.**

A Schematic overview of experimental design. C57BL/6 mice received $2 \times 10^6$ OT-I cells alone or were additionally immunised with 10,000 γ-radiation attenuated parasites intravenously.

B Flow cytometry plots show the gating strategy for identifying K$^b$-SIINFEKL$^+$ CD11a$^{hi}$ CD8$^+$ T cells.

C Temporal analysis of antigen-specific CD8$^+$ T-cell responses. Peripheral blood was obtained on days 4, 7, 14, 21, 42 and 88 post-immunisation with WT (triangles; $n = 3–9$), CSP$^{SIINFEKL}$ (orange squares; $n = 4–8$) or UIS4$^{SIINFEKL}$ (blue circles; $n = 4–10$) sporozoites, or no parasites (diamonds; $n = 2–5$) and stained for K$^b$-SIINFEKL$^+$ CD11a$^{hi}$ CD8$^+$ T cells. Line graph shows mean values (± SEM) from representative experiments. (*$P < 0.05$; **$P < 0.01$; ***$P < 0.001$; Welch's $t$-test comparing CSP$^{SIINFEKL}$ and UIS4$^{SIINFEKL}$). See Appendix Table S1 for the number of mice used per timepoint and per group. See Appendix Table S4 for exact $P$-values.

D–H C57BL/6 mice ($n = 4$ per group), which received $2 \times 10^6$ CFSE-labelled OT-I splenocytes, were immunised with 10,000 γ-radiation attenuated WT, CSP$^{SIINFEKL}$ or UIS4$^{SIINFEKL}$ sporozoites intravenously. 5 days later, mice were sacrificed, spleens harvested and splenocytes assessed for (D) CFSE dilution of CD8$^+$ T cells, (E) CFSE dilution of antigen-experienced K$^b$-SIINFEKL$^+$ CD11a$^{hi}$ CD8$^+$ T cells and stained $ex$ $vivo$ (F-H) for effector CD8$^+$ T-cell surface markers. Shown are flow cytometry plots of K$^b$-SIINFEKL co-staining with markers of effector phenotypes: (F) CD11a$^{hi}$, (G) CD62L$^{lo}$ and (H) CD49d$^{hi}$.

higher numbers (proportion and absolute numbers) of IFN-γ-secreting CD8$^+$ T cells were observed at day 14 and day 42 following immunisation with CSP$^{SIINFEKL}$ compared with UIS4$^{SIINFEKL}$ sporozoites (Fig 3E–G). In addition, these CD8$^+$ T cells also expressed TNF and IL-2 in a similar differential pattern, suggesting potential polyfunctionality (Fig EV4). Additional gating strategies for the characterisation of IFN-γ-secreting CD8$^+$ T cells are shown in Appendix Fig S2.

Altogether, even though effector and effector memory CD8$^+$ T-cell responses can be detected against both sporozoite surface protein and EEF vacuolar membrane protein antigens following immunisation with γ-radiation attenuated sporozoites, the two

antigens show a striking difference in the magnitude of CD8$^+$ T-cell responses they induce.

## Higher magnitude of endogenously produced splenic and intrahepatic antigen-specific CD8$^+$ T-cell responses to a sporozoite antigen

Previous work tracking responses to SIINFEKL-tagged proteins has used adoptively transferred cells from OT-I mice, with all T cells from these mice expressing T-cell receptors specific to SIINFEKL (Cockburn $et$ $al$, 2011; Montagna $et$ $al$, 2014). We employed this robust approach by adoptively transferring a fixed amount of OT-I

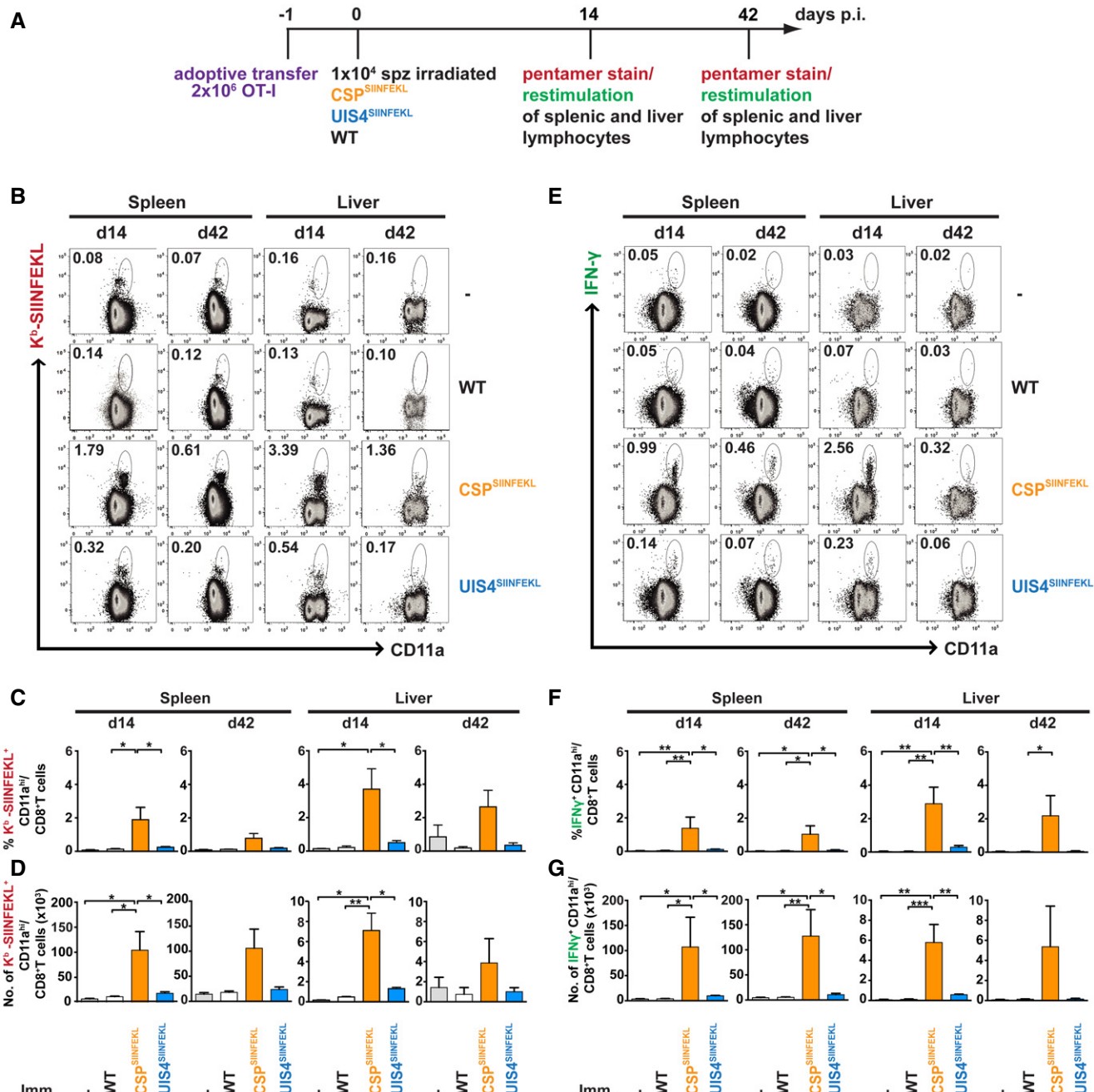

**Figure 3. Sporozoite surface antigen induces a higher CD8[+] T-cell response than EEF vacuolar membrane antigen in the spleen and liver.**

A   Schematic overview of experimental design. C57BL/6 mice received $2 \times 10^6$ OT-I cells alone ($n = 2$–4) or were additionally immunised with 10,000 γ-radiation attenuated WT ($n = 3$–6), CSP[SIINFEKL] ($n = 4$) or UIS4[SIINFEKL] ($n = 4$) sporozoites intravenously. See Appendix Table S2 and S3 for the number of mice used for pentamer staining and peptide restimulation. Spleens and livers were harvested either at day 14 or day 42.

B–G   Proportions and numbers of (B-D) K[b]-SIINFEKL[+] CD8[+] T cells were enumerated, or (E-G) IFN-γ-secreting CD8[+] T cells following restimulation *ex vivo* with SIINFEKL peptide were quantified. Flow cytometry plots show representative percentages of CD8[+] T cells co-stained with CD11a and (B) K[b]-SIINFEKL or (E) IFN-γ. The upper panel of bar charts (C, F) show the percentage of co-stained CD8[+] T cells and the lower panel (D, G) the absolute cell counts. Bar charts show mean values ($\pm$ SEM) from representative experiments (*$P < 0.05$; **$P < 0.01$; ***$P < 0.001$; one-way ANOVA with Tukey's multiple comparison test). See Appendix Table S4 for exact $P$-values.

splenocytes in order to augment the response and allow visualisation (Figs 2 and 3). Next, we wanted to explore whether we can capture the endogenous K[b]-SIINFEKL[+] CD11a[hi] CD8[+] T-cell

population, which is elicited by intravenous immunisation with γ-radiation attenuated sporozoites without OT-I cell transfer. We performed *ex vivo* restimulation of lymphocytes with SIINFEKL

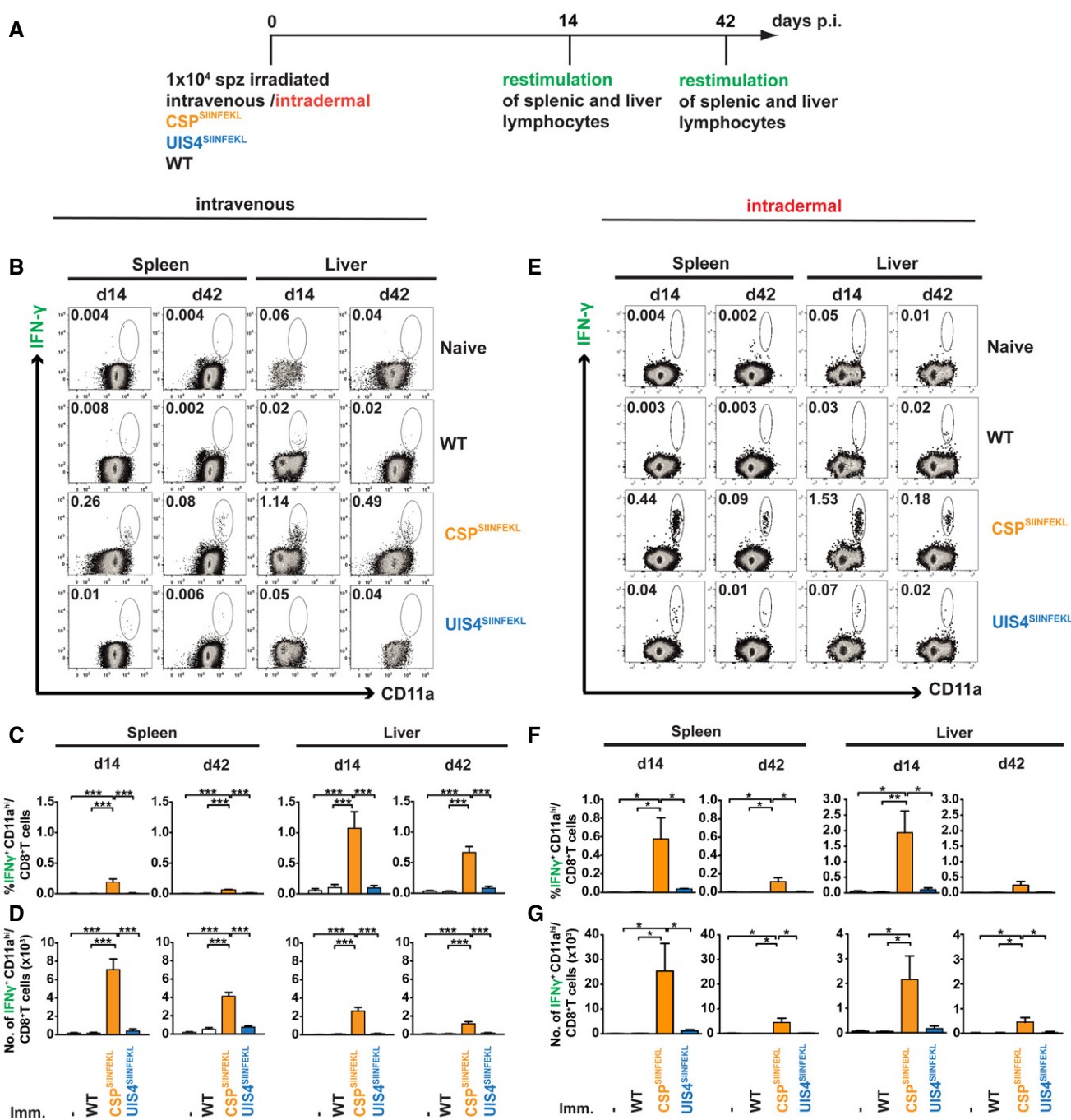

**Figure 4. Higher magnitude of endogenous SIINFEKL-specific CD8+ T-cell responses induced by a sporozoite antigen.**

A  Schematic overview of experimental design.

B–G  C57BL/6 mice received 10,000 γ-radiation attenuated WT, CSP$^{SIINFEKL}$ or UIS4$^{SIINFEKL}$ sporozoites, either (B-D) intravenously ($n = 6$–8 mice per group) or (E-G) intradermally ($n = 4$ mice per group). Additional control mice did not receive sporozoites ($n = 6$–8 intravenously, $n = 4$ intradermally). Spleens and livers were harvested either at day 14 ($n = 5$ livers and $n = 6$ spleens for intravenous immunisation) or day 42 ($n = 8$ for intravenous immunisation), and IFN-γ-secreting lymphocytes following restimulation *ex vivo* with SIINFEKL peptide were quantified. Flow cytometry plots show representative percentages of CD8+ T cells co-stained with IFN-γ and CD11a (B, E). The upper panel of bar charts (C, F) show the percentage of CD11a$^{hi}$ IFN-γ+ CD8+ T cells and the lower panel (D, G) the absolute cell counts. Bar charts show mean values ($\pm$ SEM) from representative experiments (*$P < 0.05$; **$P < 0.01$; ***$P < 0.001$; one-way ANOVA with Tukey's multiple comparison test). See Appendix Table S4 for exact $P$-values.

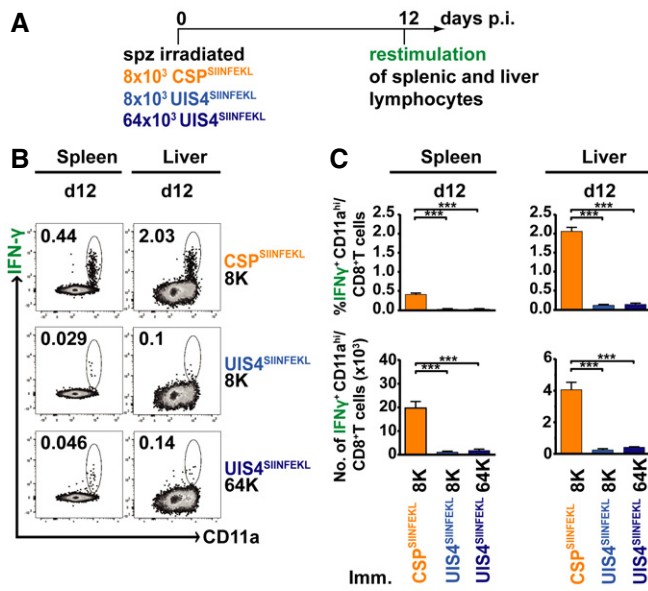

**Figure 5. Increasing antigen dose does not improve antigen-specific CD8+ T-cell responses to an EEF vacuolar membrane protein.**

A Schematic overview of experimental design. C57BL/6 mice (*n* = 4 per group) received an intravenous dose of 8,000 γ-radiation attenuated CSP^SIINFEKL or UIS4^SIINFEKL sporozoites or 64,000 γ-radiation attenuated UIS4^SIINFEKL sporozoites. Spleens and livers were harvested at day 12 and IFN-γ-secreting lymphocytes following restimulation *ex vivo* with SIINFEKL peptide were quantified.

B Flow cytometry plots show representative CD8+ T cells co-stained with IFN-γ and CD11a.

C The upper panel of bar charts show the percentage of CD11a^hi IFN-γ+ CD8+ T cells and the lower panel the absolute cell counts. Bar charts show mean values (± SEM) from representative experiments (***P < 0.001; one-way ANOVA with Tukey's multiple comparison test). See Appendix Table S4 for exact *P*-values.

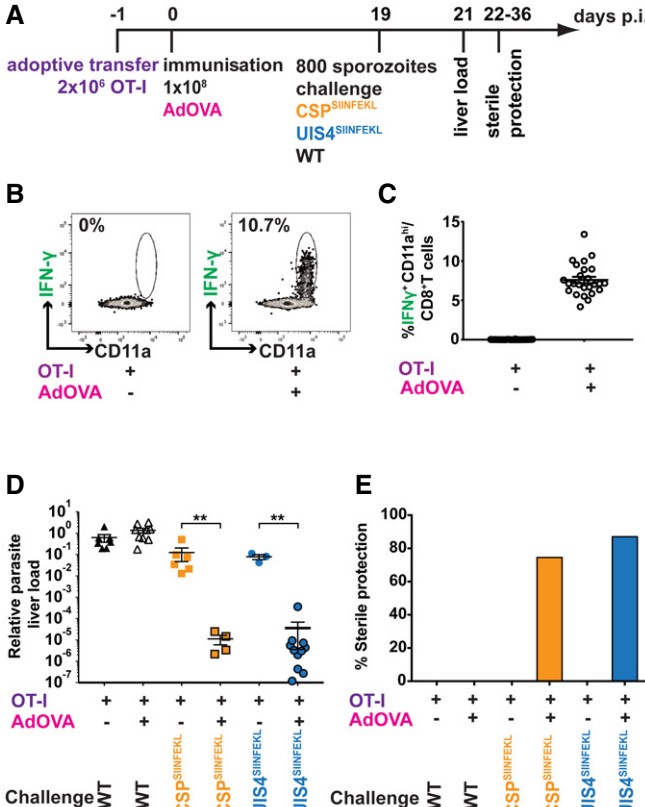

**Figure 6. Both sporozoite surface and EEF vacuolar membrane antigens are susceptible to vaccine-induced CD8+ T cells for killing, resulting in sterile protection.**

A Schematic overview of experimental design. Mice received 2 × 10^6 OT-I splenocytes, and 1 day later, they were either vaccinated with 1 × 10^8 ifu recombinant AdHu5 expressing whole ovalbumin (AdOVA) (*n* = 5 per challenge group) or left untreated (*n* = 4 per challenge group).

B, C Flow cytometry (B) and scatter plots (C) represent CD8+ T cells derived from peripheral blood co-stained with IFN-γ and CD11a, following *ex vivo* restimulation with SIINFEKL.

D Protective efficacy as measured by quantitative real-time PCR. Groups of mice were vaccinated as described and challenged 19 days later with 10,000 WT (*n* = 7 non-vaccinated, *n* = 9 vaccinated), CSP^SIINFEKL (*n* = 6 non-vaccinated, *n* = 4 vaccinated) or UIS4^SIINFEKL sporozoites (*n* = 3 non-vaccinated, *n* = 11 vaccinated). 42 h later livers were removed, and parasite load was assessed by qPCR. Plots show the relative parasite load of mice in each condition (**P < 0.01; Mann–Whitney *U*-test). See Appendix Table S4 for exact *P*-values.

E Proportion of sterile protection after immunisation. Mice (vaccinated group, *n* = 8; control group, *n* = 4) were vaccinated as described and were challenged with 800 WT, CSP^SIINFEKL or UIS4^SIINFEKL sporozoites. Daily blood smears were taken from day 3 to 14 post-challenge to check for parasitaemia.

Data information: Data for B-E are representative of two experiments performed with scatter plots showing mean values (± SEM).

peptide followed by flow cytometry and were able to clearly identify the endogenous population with a trend complementary to our earlier results (Fig 4A–D). Immunisation with CSP^SIINFEKL sporozoites elicited a superior splenic and liver CD8+ T-cell response than with UIS4^SIINFEKL sporozoites. As expected, the proportion and absolute cell numbers were considerably lower than with adoptive transfer of OT-I cells, but this did not preclude the ability to detect IFN-γ-secreting CD8+ T cells and capture the differences between the two groups.

Under normal conditions of transmission, sporozoites are delivered into the host skin by mosquito bite. All preceding immunisation experiments were performed with parasites injected intravenously. As a proxy for the natural route of infection, whilst ensuring consistent quantities of parasites were inoculated, γ-radiation-attenuated CSP^SIINFEKL and UIS4^SIINFEKL sporozoites were injected via the intradermal route into the ear pinnae. Under these conditions, CSP still induced a greater number of IFN-γ-secreting SIINFEKL-specific CD8+ T cells following restimulation with SIIN-FEKL compared with UIS4, with a comparable magnitude as after intravenous injection (Fig 4E–G; Appendix Fig S3A). Thus, these biologically and immunologically more appropriate data entirely recapitulate the strong immunogenicity of a sporozoite surface antigen compared with an EEF vacuolar membrane protein.

## Increasing the amount of EEF vacuolar membrane antigen does not impact its immunogenicity

CSP and UIS4 are critical proteins expressed by the sporozoite and EEF, respectively, and both proteins are important for survival and succession into the subsequent life stage and parasite form (Mueller

et al, 2005; Coppi et al, 2011; Silvie et al, 2014). Previous studies have shown that the magnitude of the CD8[+] T-cell response to a sporozoite surface antigen depended on the amount of parasites used for immunisation (Hafalla et al, 2002). Hence, poor immunogenicity of an EEF vacuolar membrane protein could be a result of the lower level of protein expression during parasite infection. It is possible to enhance CD8[+] T-cell responses by increasing the number of parasites used for immunisation (Hafalla et al, 2002). Therefore, we immunised groups of mice with 8,000 γ-radiation-attenuated CSP[SIINFEKL], 8,000 UIS4[SIINFEKL] or 64,000 UIS4[SIINFEKL] sporozoites and compared the magnitude of the elicited antigen-specific responses. Strikingly, the CD8[+] T-cell response following 8× sporozoite immunisation dose with UIS4[SIINFEKL] did not increase proportionally and was not significantly higher than immunisation with a 1× dose (Fig 5A–C; Appendix Fig S3B). This result suggests that, in the context of attenuated sporozoite immunisation, EEF vacuolar membrane antigens are poorly immunogenic and increasing antigen fails to substantially improve the magnitude of CD8[+] T-cell responses.

### Immunogenicity of parasite antigens does not predict effector responses following vaccination

Our findings thus far showed that sporozoite surface proteins appear more immunogenic than EEF vacuolar membrane proteins and raised an intriguing and important question; does immunogenicity predict susceptibility to vaccine-induced effector responses? To address this, we vaccinated mice, which had received OT-I cells, with a recombinant adenovirus expressing full-length ovalbumin (de Cassan et al, 2011) (Fig 6A). This vaccination protocol resulted in frequencies of ~7.5% SIINFEKL-specific CD8[+] T cells in peripheral blood (Fig 6B and C; Appendix Fig S3C). Vaccinated mice were then challenged with live CSP[SIINFEKL] or UIS4[SIINFEKL] sporozoites, and protection was assessed 19 days after vaccination by two complementary assays; (i) determination of the reduction of parasite load in the liver 42 h after sporozoite challenge (Fig 6D) and (ii) induction of sterile protection (Fig 6E). Vaccinated mice challenged with CSP[SIINFEKL] or UIS4[SIINFEKL] sporozoites showed a dramatic reduction in parasite load in the liver (Fig 6D) as compared to vaccinated mice challenged with WT parasites. Strikingly, contrary to the differential CD8[+] T-cell responses induced by CSP and UIS4, there was no statistical difference in the protection observed when vaccinated mice were challenged with either CSP[SIINFEKL] or UIS4[SIINFEKL] sporozoites. Consistent with these findings, both groups of vaccinated mice challenged with either CSP[SIINFEKL] or UIS4[SIINFEKL] sporozoites exhibited sterile protection of comparable levels (Fig 6E; Table 1) determined by microscopic analysis of blood smears post-challenge. These findings indicate that spatial and temporal aspects of antigen expression

may affect protein immunogenicity in the context of parasitic infection but not necessarily the same target's susceptibility for antigen-specific CD8[+] T-cell killing.

## Discussion

The malaria pre-erythrocytic stages have been a prime target for the development of a Pf vaccine for more than 35 years. Indeed, RTS,S/AS01, the most advanced malaria sub-unit vaccine candidate to date, is based on PfCSP, the major surface protein of sporozoites (Cohen et al, 2010). Yet, final results of the Phase III trial showed that RTS,S/AS01 offers only modest efficacy, which rapidly wanes over time (Rts SCTP, 2015). Thus, there is an imperative need not only to widen the pursuit for new sub-unit vaccine candidates, but also to radically improve the antigen selection process. Antigens are generally prioritised based on a range of criteria, including their immunogenicity in the context of parasitic infection. We examined this notion in an infection and vaccination model for Pb pre-erythrocytic stages. We evaluated whether antigen immunogenicity and accessibility are decisive features of vaccine design.

The malaria pre-erythrocytic stages consist of two spatially different parasite forms: extracellular sporozoites and intracellular EEFs. The transformation of sporozoites into EEFs involves regulation at both transcriptional (Iwanaga et al, 2012) and translational (Muller et al, 2011; Zhang et al, 2016) levels, resulting in divergent spatial and temporal expression of many antigens that are distinct for each parasite form (Tarun et al, 2008). Whilst our current understanding of immune responses to malaria pre-erythrocytic stages has focussed on CSP, the lack of well-defined epitopes that are expressed only by EEFs has restrained fundamental studies investigating the contributions of EEF antigens in parasite-induced CD8[+] T-cell responses and their value as target of vaccines.

In this study, we contrasted the development of CD8[+] T-cell responses induced by CSP and UIS4, two major proteins expressed by sporozoites and EEFs, respectively. We generated transgenic Pb parasites where SIINFEKL is expressed as part of either CSP or UIS4, allowing the presentation of the epitope at the same space and time as the respective protein. This approach is in contrast to a more common strategy of expressing the whole, or a fragment of, ovalbumin inserted as a transgene into the Pb genome, and then tracking the immune response elicited by an extraneous molecule (Lin et al, 2014; Montagna et al, 2014). Since CSP is expressed in both sporozoites and EEFs, the processing and presentation of the SIINFEKL in CSP[SIINFEKL] occurs as soon as sporozoites are inoculated and are able to interact with CD11c[+] dendritic cells, which cross-present antigens via an endosome-to-cytosol pathway (Cockburn et al, 2011); CSP also has direct access to the hepatocyte's cytosol for

**Table 1. Prepatency of AdOVA immunised and challenged mice.[a]**

| Challenge | WT | WT | CSP[SIINFEKL] | CSP[SIINFEKL] | UIS4[SIINFEKL] | UIS4[SIINFEKL] |
|---|---|---|---|---|---|---|
| AdOVA Immunisation | − | + | − | + | − | + |
| Infected / Challenged | 4 / 4 | 8 / 8 | 4 / 4 | 2 / 8 | 4 / 4 | 1 / 8 |
| Prepatency (days) | 3.75 | 4 | 3.35 | (5)[b] | 3.5 | (5)[b] |

[a]All mice received 2 × 10[6] OT-1 cells 1 day prior to immunisation.
[b]Brackets are indicative of the small proportion of mice that became blood stage-positive at the day indicated, while all others remained negative until day 14.

processing and presentation of the CSP-derived epitope (Cockburn et al, 2011). Since UIS4 is expressed only in the PVM of EEFs, processing and presentation of the epitope in UIS4$^{SIINFEKL}$ are restricted to just hepatocytes, although a recent report has suggested a role for monocyte-derived CD11c$^+$ cells in acquiring parasites during EEF stages (Kurup et al, 2019).

Our results establish that following sporozoite immunisation, a sporozoite surface protein induces superior CD8$^+$ T-cell responses—as measured both by pentamer staining and by IFN-γ secretion following peptide stimulation—than an EEF vacuolar membrane protein. Detailed kinetic and phenotypic analysis of the development of antigen-specific CD8$^+$ T cells to both CSP and UIS4 revealed that the responses differ in magnitude, demonstrating the ability of both antigens to elicit effector and effector memory responses. There was no difference in our results whether sporozoites are delivered using the commonly used intravenous immunisation or the more physiological intradermal route. We also showed that increasing the number of UIS4$^{SIINFEKL}$ parasites used for immunisation did not augment CD8$^+$ T-cell responses, signifying that the poor immunogenicity of an EEF vacuolar membrane protein is not due to the level of UIS4 expression during parasite infection. Our findings support the idea that EEF antigens have minimal contributions to the magnitude of immune responses following whole sporozoite immunisation, which corroborates with prior data showing that hepatocytes are poor at priming T-cell responses (Thomson & Knolle, 2010; Bertolino & Bowen, 2015).

In our initial experiments, mice received cells from transgenic OT-I mice, which express SIINFEKL-specific TCRs on their CD8$^+$ T cells, prior to sporozoite immunisation. The results from these experiments were compared with experimental set-ups, where mice did not receive OT-I cells; the results of the latter solely quantified the more physiological endogenous CD8$^+$ T-cell response. As expected, mice that received transgenic CD8$^+$ T cells elicited higher CD8$^+$ T-cell responses—of one order of magnitude—as compared to endogenous responses. Nonetheless, the trend of frequencies is consistent in establishing that a sporozoite surface antigen induces a higher CD8$^+$ T-cell response than EEF vacuolar membrane antigen. It is of fundamental interest to understand further the fine specificities (i.e. proliferative capacity, phenotypic and functional characteristics) and contributions of the transferred versus endogenous CD8$^+$ T cells in the resulting immune response. Further investigations using congenic mice are warranted to address these limitations.

CD8$^+$ T cells recognise peptides directly processed and presented by parasitised hepatocytes to exert their functions (Bongfen et al, 2007; Chakravarty et al, 2007; Cockburn et al, 2011; Balam et al, 2012; Huang et al, 2015). Thus, we also evaluated whether sporozoite or EEF-derived epitopes are accessible, and therefore susceptible to vaccine-induced effector CD8$^+$ T cells. Regardless of their differing immunogenicities in the context of parasitic infection, we demonstrated that both sporozoite and EEF antigens are effectively targeted by antigen-specific effector CD8$^+$ T cells, which were generated by vaccinating mice that received OT-I cells with a recombinant adenovirus expressing the cognate epitope. The transfer of OT-I ensured the induction of high levels of vaccine-induced effector CD8$^+$ T cells in a single vaccination, with the ensuing frequencies comparable to those obtained by prime-boost vaccinations with recombinant adeno- followed by

vaccinia viruses (Bruna-Romero et al, 2001; Gilbert et al, 2002; Reyes-Sandoval et al, 2010; Reyes-Sandoval et al, 2011), or with peptide-loaded dendritic cells followed by recombinant Listeria (Doll et al, 2016). These prime-boost strategies have consistently shown to induce high numbers of antigen-specific CD8$^+$ T cells (Bruna-Romero et al, 2001; Gilbert et al, 2002; Reyes-Sandoval et al, 2010; Reyes-Sandoval et al, 2011; Doll et al, 2016) necessary for protection (Schmidt et al, 2010). Previous work in mouse models and CSP-based adenovirus vaccines showed the generation of endogenous responses that yielded only up to 40% sterile immunity, despite two orders of magnitude in the reduction in parasite load in the liver after challenge with viable sporozoites (Rodrigues et al, 1997; Rodrigues et al, 1998). Importantly, using the OT-I transfer model, our study shows that mice harbouring similarly high levels of vaccine-induced, antigen-specific CD8$^+$ T cells were comparably protected when challenged with either CSP$^{SIINFEKL}$ or UIS4$^{SIINFEKL}$. These findings imply that both sporozoite and EEF antigens comparably access the antigen presentation pathways in hepatocytes leading to recognition of defined epitopes. Our study reiterates that high levels of vaccine-induced, antigen-specific CD8$^+$ T cells are needed to achieve protection in mice, and likely in humans. Our work provides proof-of-concept for vaccines targeting the complete malaria pre-erythrocytic stages. The translational challenge will now be to design vaccine formulations, which evoke and maintain high levels of antigen-specific human CD8$^+$ T cells either given as single dose or as part of a prime-boost approach.

Our study is the first demonstration that poor natural immunogenicity, in this case of an EEF antigen, does not preclude antigen-specific CD8$^+$ T-cell killing. Our findings that antigen immunogenicity in this context is an inadequate predictor of vaccine efficacy have wide-ranging translational implications on antigen prioritisation for the design and testing of next-generation pre-erythrocytic Pf vaccines. Thus, the strategy to screen for T-cell responses in naturally infected or sporozoite-immune human volunteers to pre-clinically prioritise vaccine candidates requires some form of reassessment. Conventional immunological assays, aimed at identifying highly immunogenic antigens, may fail to discover those candidates with the potential to induce superior levels of protective immunity. For this proof-of-concept study, we selected CSP and UIS4 as the best characterised representatives of sporozoite and EEF vacuolar antigens. It will be interesting to systematically compare global allelic diversity of sporozoite and EEF antigens. The observation of allele-specific RTS,S/AS01 vaccine efficacy and low proportions of vaccine-matched CSP alleles in sub-Saharan Africa (Neafsey et al, 2015) underscores the potential critical advantage of EEF antigens, which likely display only moderate diversity across P. falciparum strains vis-à-vis extended diversity of sporozoite surface antigens. It is noteworthy that for other stages of malaria infection, conserved antigens that give limited or no responses, e.g. PfRH5 (Douglas et al, 2011; Osier et al, 2014) and sexual stage antigens (Kapulu et al, 2015), are promising antibody targets for malaria vaccines under clinical development. Together, our study establishes that EEF-expressed antigens are prospective vaccine targets and thus effectively increases the pool of existing pre-erythrocytic stage vaccine targets for the malaria community to explore and translate to clinical trials.

A key direction for future research will be finding new assays to easily distinguish and validate promising vaccine targets, namely those antigens that can protect, via susceptibility to vaccine-induced $CD8^+$ T cells, rather than those that naturally induce strongly immunogenic responses. In addition, our findings open new questions as to whether EEF-expressed antigens are optimal antigens for the generation of endogenous liver-resident memory ($T_{RM}$) $CD8^+$ T cells. $T_{RM}$ cells were recently identified to persist in and continuously perambulate the liver to evoke immune responses following antigen challenge (Fernandez-Ruiz et al, 2016; Valencia-Hernandez et al, 2020). Further work, utilising congenic mice, is warranted to characterise in detail the intrahepatic cells induced by $CSP^{SIINFEKL}$ and $UIS4^{SIINFEKL}$ parasites. There is current translational interest in exploiting $T_{RM}$ cells for vaccine strategies against malaria pre-erythrocytic stages. Current pre-clinical approaches are aimed at initially activating T cells in the spleen by a priming vaccine, and another delivery system is used to trap these cells in the liver to form $T_{RM}$ cells. It is conceivable that EEF-expressed antigens could already prime $T_{RM}$ cells, which can be boosted by novel vaccine delivery systems.

Ultimately, the molecular mechanisms of presentation of EEF antigens, those expressed in the PVM and within the parasite itself, onto the surface of infected hepatocytes remains to be fully understood. Determination of the processes involved in parasite antigen presentation in the pre-erythrocytic stages of malaria may elucidate links to protection and the identification of hitherto neglected antigens that could drive the development of a highly efficacious malaria vaccine.

# Materials and Methods

## Reagents and Tools table

| Reagent | Source | Identifier |
| --- | --- | --- |
| **Antibodies** | | |
| Anti-mouse CD3e-V500 (500A2) | BD Biosciences | Cat# 560771, RRID: AB_1937314 |
| Anti-mouse CD8a-PerCP-Cy5.5 (53-6.7) | Thermo Fisher Scientific | Cat# 45-0081-82, RRID: AB_1107004 |
| Anti-mouse CD11a-e450 (M17/4) | Thermo Fisher Scientific | Cat#48-0111-82, RRID: AB_11064445 |
| Anti-mouse CD11a-FITC (M17/4) | Thermo Fisher Scientific | Cat#11-0111-82, RRID: AB_464931 |
| Anti-mouse CD44-e450 (IM7) | Thermo Fisher Scientific | Cat# 48-0441-82, RRID: AB_1272246 |
| Anti-mouse CD49d-PE (R1-2) | Thermo Fisher Scientific | Cat# 12-0492-82, RRID: AB_465697 |
| Anti-mouse CD62L-PE-Cy7 (MEL-14) | Thermo Fisher Scientific | Cat# 25-0621-82, RRID: AB_469633 |
| Anti-mouse IFN-γ-APC (XMG1.2) | Thermo Fisher Scientific | Cat# 17-7311-82, RRID: AB_469504 |
| Anti-mouse TNF-α-FITC (MP6-XT22) | Thermo Fisher Scientific | Cat# 11-7321-82, RRID: AB_465418 |
| Anti-mouse IL-2-PE-Cy7 (JES6-5H4) | Thermo Fisher Scientific | Cat# 25-7021-82, RRID: AB_1235004 |
| APC-labelled Pro5 MHC pentamer H-2-$K^b$-SIINFEKL | ProImmune | Customised |
| Anti-mouse CD45.2-Alexa647 (104) | Biolegend | Cat# 109818, RRID:AB_492870 |
| Mouse anti-*Plasmodium berghei* CSP | Potocnjak et al, 1980; PMID: 6991628 | RRID:AB_2650479 |
| Rabbit polyclonal anti-*Plasmodium berghei* UIS4 | Muller et al, 2011: PMID: 21673790 | N/A |
| Mouse anti-*Plasmodium berghei* HSP70 | Tsuji et al, 1994: PMID: 8153120 | RRID:AB_2650482 |
| Alexa Fluor 488 goat anti-mouse | Thermo Fisher Scientific | Cat # A-11029 RRID:AB_2534088 |
| Alexa Fluor 546 goat anti-rabbit | Thermo Fisher Scientific | Cat # A-11010 RRID:AB_2534077 |
| **Experimental Models: Cell Lines** | | |
| Huh-7 | Human hepatoma cells | Nakabayashi, H., Taketa, K., Miyano, K., Yamane, T. & Sato, J. Growth of human hepatoma cells lines with differentiated functions in chemically defined medium. Cancer Res. 42, 3858–3863 (1982). |
| **Experimental Models: Organisms/Strains** | | |
| C57BL/6J mice | Charles River or LSHTM | RRID:IMSR_JAX:000664 |
| CD-1 mice | LSHTM | RRID:IMSR_CRL:22 |
| NMRI mice | Charles River | RRID:IMSR_CRL:605 |
| OT-I mice | Charles River | RRID:IMSR_CRL:642 |
| *Anopheles stephensi* SK | (Feldmann and Ponnudurai, 1989) | |

## Methods and Protocols

### Ethics and animal experimentation

Animal procedures were performed in accordance with the German "Tierschutzgesetz in der Fassung vom 18. Mai 2006 (BGBl. I S. 1207)" which implements the directive 2010/63/EU from the European Union. Animal experiments at London School of Hygiene and Tropical Medicine were conducted under licence from the United Kingdom Home Office under the Animals (Scientific Procedures) Act 1986. All animal experiments were carried out under institutional care protocols. NMRI, CD-1, C57BL/6 and OT-I laboratory mouse strains were bred in house at LSHTM or purchased from Charles River Laboratories (Margate, UK or Sulzfeld, Germany). Female mice were used for experiments at the age of 6–8 weeks. Mice were placed into experimental cages of the required sample size by animal staff and not investigators; thus, an element of random group allocation occurred. Where more animals were required for a particular cage, selected animals were chosen at random for relocation. No blinding was performed. Sample size was estimated based on our previous experience and mouse availability. This project is in compliance with the ARRIVE guidelines.

### Generation of transgenic parasites

Transgenic *P. berghei* ANKA mutants CSP[SIINFEKL] and UIS4[SIINFEKL] were developed using double homologous recombination. In the CSP[SIINFEKL] mutant, the CSP gene is altered so the epitope SYIPSAEKI (residues 252-260) is replaced with the H-2[b] restricted *Gallus gallus* ovalbumin epitope SIINFEKL. In the UIS4[SIINFEKL] mutant, the SIINFEKL epitope is appended to the C-terminal end of the UIS4 protein. As such, B3D-CSP[SIINFEKL] plasmid was assembled by successive cloning of three fragments, CSP-C, CSP-B and CSP-A, obtained by PCR amplification from *P. berghei* ANKA genomic DNA followed by restriction enzyme digestion. These fragments correspond, respectively, to a 3′ homology region downstream of CSP (CSP-C, 0.7 kb), a fragment comprising the CSP ORF downstream of the SYIPSAEKI epitope followed by the CSP 3′ UTR (CSP-B, 0.8 kb) and a fragment comprising a 5′ promoter region followed by the CSP-modified ORF where the SYIPSAEK coding sequence has been replaced by a SIINFEKL coding sequence (CSP-A, 1.8 kb). The resulting B3D-CSP[SIINFEKL] plasmid, containing the *Toxoplasma gondii* dihydrofolate reductase/thymidylate synthase (*TgDHFR/TS*) pyrimethamine resistance cassette flanked by CSP-A and CSP-B on one side, and CSP-C on the other, was linearised with *Not*I and *Sac*II before transfection. Integration of the construct after double crossover homologous recombination results in replacement of the WT CSP gene by a modified copy containing the SIINFEKL coding sequence instead of the SYIPSAEKI coding sequence. The B3D-UIS4[SIINFEKL] plasmid was assembled by successive cloning of three fragments, UIS4-A, UIS4-B and UIS4-C, obtained by PCR amplification from *P. berghei* ANKA genomic DNA followed by restriction enzyme digestion. These fragments correspond, respectively, to a fragment comprising a 5′ upstream sequence followed by the UIS4 entire ORF fused in frame to the SIINFEKL coding sequence (UIS4-A, 1.2 kb), to the UIS4 3′ UTR sequence (UIS4-B, 0.6 kb) and to a 3′ homology region downstream of UIS4 (UIS4-C, 0.9 kb). The resulting B3D-UIS4[SIINFEKL] plasmid, containing the *TgDHFR/TS* pyrimethamine resistance cassette flanked by UIS4-A and UIS4-B on one side,

and UIS4-C on the other, was linearised with *Sac*II and *Kpn*I before transfection. Integration of the construct after double crossover homologous recombination results in replacement of the WT UIS4 gene by a modified copy containing the SIINFEKL coding sequence just upstream of a STOP codon. *P. berghei* CSP[SIINFEKL] and UIS4[SIINFEKL] parasites were generated by transfection of *P. berghei* ANKA with linearised B3D-CSP[SIINFEKL] and B3D-UIS4[SIINFEKL] plasmids, respectively. Purified schizonts of WT *P. berghei* ANKA (clone c15cy1) were transfected with 5–10 μg of linearised plasmid by electroporation using the AMAXA Nucleofector™ device (program U33), as described (Janse *et al*, 2006), and immediately injected intravenously in the tail vein of a mouse. The day after transfection, pyrimethamine (70 mg/l) was administrated in the mouse drinking water, for selection of transgenic parasites. Transgenic clones were isolated after limiting dilution and injection into mice. Correct integration of the constructs and purity of the transgenic lines was verified by analytical PCR using primer combinations specific for the unmodified CSP or UIS4 locus, and for the 5′ and 3′ recombination events. All primers used in this study are indicated in Table 2.

### Plasmodium berghei ANKA immunisation

*Plasmodium berghei* WT (strain ANKA clone c15cy1 or clone 507) parasites and CSP[SIINFEKL] and UIS4[SIINFEKL] (clone c15cy1) parasites were maintained by continuous cycling between murine hosts (NMRI or CD-1) and *Anopheles stephensi* mosquitoes. Infected mosquitoes were kept in incubators (Panasonic and Mytron) at 80% humidity and 20°C. Sporozoites were isolated from salivary glands and γ-irradiated at $1.2 \times 10^4$ cGy. Mice were immunised intravenously in the lateral tail vein or intradermally in the ear pinnae with 10,000 sporozoites, unless otherwise stated, and challenged with either 1,000 or 10,000 sporozoites injected intravenously.

### Indirect fluorescent antibody staining (IFA) of sporozoites

Epoxy-covered 8-well glass slides were coated with 3% BSA-RPMI. 10,000 sporozoites were added per well in 3% BSA-RPMI and incubated for 45 min during which the shed surface proteins are deposited in the gliding motility process. Sporozoites and their trails were stained with a mouse anti-CSP (Potocnjak *et al*, 1980) (1:300) primary antibody and a rabbit polyclonal anti-*Pb*UIS4 (Muller *et al*, 2011) (1:500) primary antibody and the respective fluorescently labelled secondary antibodies (1:2,000). Nuclei were stained with Hoechst 33342 (1:2,000) and slides mounted with "Fluoromount-G" (Southern Biotech). Sporozoites and trails were analysed by fluorescent microscopy (Zeiss Axio Observer).

### In vitro infection of hepatoma cells and fluorescent staining

*In vitro* EEF development was analysed in infected Huh7 hepatoma cells for 24 and 48 h. Triplicate Labtek (Permanox plastic—Nunc) wells were infected with 10,000 transgenic CSP[SIINFEKL] or UIS4[SIINFEKL] parasites and duplicate wells were infected with 10,000 WT parasites. Infected cells were analysed by fluorescence microscopy using a mouse anti-*Pb*HSP70 (Tsuji *et al*, 1994) (1:300) and a rabbit polyclonal anti-*Pb*UIS4 (Muller *et al*, 2011) (1:500) primary antibody, the respective fluorescently labelled secondary antibodies (1:2,000) and nuclear staining with Hoechst 33342 (1:2,000). Stainings were analysed by fluorescent microscopy (Zeiss Axio Observer).

**Table 2. Primers used to generate plasmids and genotype parasites.**

|  | Oligonucleotide | Sequence 5′ 3′ |
|---|---|---|
| Production of B3D-CSP^SIINFEKL construct | CSP-A forward | ATAAGAATGCGGCCGCATGGTTATATTTTGTGCAATGCTAAAATGG |
|  | CSP-A reverse | CGGAATTCTAGTATCAGTTTTTCAAAGTTGATTATACTATCGTCATTATTATTATTTTTGTTATTG |
|  | CSP-B forward | GGACTAGTGAATTCGTTAAACAGATCAGGGATAGTATCACAGAGG |
|  | CSP-B reverse | CCGCAATTGTACAAAAAATATTTTCGACAAAGGATAACG |
|  | CSP-C forward | CCCAAGCTTTGGGAATCTATTTTACAATATTATTTAAGGG |
|  | CSP-C reverse | CGGGGTACCCCGCGGTTATTGAAAAAGACACAAAATAGCTAG |
| Production of B3D-UIS4^SIINFEKL construct | UIS4-A forward | TCCCCGCGGATAGCTATATTTTATGGTTGATCCTTTCC |
|  | UIS4-A reverse | GGACTAGTTTACAGTTTTTCAAAGTTGATTATACTTATGTATGGGCCGAATGATTTATTTTCC |
|  | UIS4-B forward | GGACTAGTTTCATTATGAGTAGTGTAATTCAGAAAGAG |
|  | UIS4-B reverse | CCGGAATTCTATGTAAAAAAGTTTGCATATACGGCTG |
|  | UIS4-C forward | CCCAAGCTTAGTGAAATATAAATATGAATGGAAGCAGCC |
|  | UIS4-C reverse | CGGGGTACCAGCAGCTAATGTCAATATATTTTATGCAC |
| Genotying of transgenic parasites | TgDHFR forward | CGCATTATATGAGTTCATTTTACACAATCC |
|  | OVA reverse | CTAGTTTACAGTTTTTCAAAGTTGATTATAC |
|  | CSP WT forward | TGTGAACTTTTCCTTATTTATTACGATTATG |
|  | CSP test forward | AATATGAGCACGCTTTTACTTTGTCCAGG |
|  | CSP test reverse | ACGAATCGAAATAAGTTACTATTCGTGCC |
|  | UIS4 test forward | TGGTTCTTAATATTATTTTGGATACATGC |
|  | UIS4 test reverse | CTCGTGTCCTTTGTAGTAAAAATAAACC |

Restriction sites in the primer sequences are underlined.

### Quantification of SIINFEKL-specific CD8+ T-cell responses

Spleens and livers were harvested from immunised or naïve mice and perfused with PBS. Lymphocytes were derived from spleens by passing through 40 or 70 μm cell strainers (Corning) and from 1× PBS-perfused livers by passing through 70 μm cell strainers (Corning). Hepatocytes were removed from homogenised livers using a modified 35% Percoll gradient (Goossens *et al*, 1990) (prepared using ratio of 1.6 10× PBS (Gibco): 12.4 Percoll (GE Healthcare) : 24.1 incomplete RPMI (Gibco)). Red blood cells were lysed with PharmLyse (BD), and lymphocytes were resuspended in complete RPMI (cRPMI- RPMI + 10% FCS + 2% Penicillin-Streptomycin + 1% L-glutamine (Gibco)). For cell counting, lymphocytes were diluted 40× with Trypan Blue (Thermo Fisher Scientific) and enumerated using a Neubauer "Improved" haemocytometer (Biochrom). Alternatively, lymphocytes were counted using a MACSQuant flow cytometer (Miltenyi Biotec), using propidium iodide (PI) (1:1,250) (Sigma-Aldrich) or, in the case of intrahepatic lymphocytes, using CD45.2-Alexa647 (1:400) (Biolegend) to distinguish between hepatocytes and lymphocytes, prior to PI administration and counting. Peripheral blood was acquired by tail vein puncture collected in Na+ heparin capillary tubes (Brand) and assayed in 96-well flat bottom plates (Corning). For CD8+ T-cell stimulations, 2–3 × 10^6 splenocytes or 1–2 × 10^5 liver cells were incubated with SIINFEKL peptide (Peptides and Elephants, Henningsdorf) at a final concentration of 10 μg/ml in the presence of Brefeldin A (eBioScience). Cells were incubated at 37°C, 5% CO$_2$ for 5–6 h, before incubation at 4°C overnight. For staining of cell surface markers and intracellular cytokines, cells were incubated for 1 h at 4°C. Cells derived from the spleen or liver were fixed with 4% paraformaldehyde, and cells from peripheral blood were fixed with 1% paraformaldehyde between the extra- and intracellular staining steps. Data were acquired by flow cytometry using an LSRII or LSRFortessa (BD). Antibodies used for staining were as follows; BD: CD3 (500A2) (1:300); eBioScience: CD8 (53-6.7) (1:225), CD11a (M17/4) (1:300), CD49d (R1-2) (1:225), CD62L (MEL-14) (1:250), CD44 (IM7) (1:250), IFN-γ (XMG1.2) (1:225), TNF (MP6-XT22) (1:250) and IL-2 (JES6-5H4) (1:250); ProImmune: H-2-K^b-SIINFEKL pentamers (3 μl/test).

### CFSE labelling of OT-I cells

Spleens from OT-I mice were lysed and cells washed twice in PBS without serum. Splenocytes resuspended at a density of 5 × 10^6 cells/ml in PBS had 1:5,000 CFSE (Thermo Fisher Scientific) added and were incubated in the dark at room temperature, with gentle inversion for 4 min. The labelling reaction was quenched with cRPMI and cells washed twice in cRPMI. Cells were recounted and 2 × 10^6 cells were injected per mouse.

### Vaccination with OVA expressing recombinant adenovirus

To assess parasite liver load after vaccination with virus-expressed OVA, groups of C57BL/6 mice were immunised with recombinant human adenovirus serotype 5 (AdHu5) expressing full-length chicken ovalbumin (de Cassan *et al*, 2011). Each mouse received 1 × 10^8 infective units (ifu) in a volume of 100 μl administered intramuscularly (50 μl into each thigh). One day prior, mice received OT-I splenocytes intravenously (2 × 10^6 cells/mouse). 19 days after vaccination, vaccinated and control mice were challenged with 10,000 WT, CSP^SIINFEKL or UIS4^SIINFEKL sporozoites administered

### The paper explained

**Problem**
Antigen selection is critical for malaria vaccine discovery. To date, sporozoite antigens are prioritised according to their immunogenicity, accessibility and capacity to elicit inhibitory antibodies, and, to a limited extent, cellular responses. Very little is known about the vaccine potential of *Plasmodium* antigens expressed during liver infection.

**Results**
We have demonstrated in a mouse malaria model that despite having poor immunogenicity, antigens expressed during liver infection are excellent targets of vaccine-induced protection through cellular immunity.

**Impact**
Our data provide a rationale for systematic evaluation of previously unrecognised parasite-derived antigens for malaria vaccine discovery and pre-clinical evaluation in formulations aimed at eliciting strong cell-mediated immunity.

intravenously. 42 h after the challenge the livers were harvested and homogenised in TRIzol (Thermo Fisher Scientific) for total RNA isolation. Afterwards, cDNA was generated using the RETROScript Kit (Ambion). Quantitative real-time PCR was performed using the StepOnePlus Real-Time PCR System and Power SYBR Green PCR Master Mix (Applied Biosystems). Relative liver parasite levels were quantified using the ΔΔCt method comparing levels of *P. berghei 18S* rRNA normalised to mouse *GAPDH* mRNA (Muller *et al,* 2011). To assess sterile protection, AdHu5 OVA-vaccinated and control mice received $2 \times 10^6$ OT-I splenocytes 1 day prior to vaccination. 14 days later, all mice were challenged with 800 WT, CSP[SIINFEKL] or UIS4[SIINFEKL] sporozoites. Blood smears were taken from day 3 to 14 after challenge to determine the presence of blood stage parasites.

### Statistics

Data were analysed using FlowJo version 9.5.3 (Tree Star Inc., Oregon, USA), Microsoft Excel and GraphPad Prism v9 (GraphPad Software Inc., CA, USA). We used Mann–Whitney *U*-test for analysing data that were not normally distributed and Welch's *t*-test or one-way ANOVA with Tukey's multiple comparison test for normally distributed data. Normal distribution was assessed using Shapiro–Wilk test.

## Data availability

This study includes no data deposited in external repositories.

**Expanded View** for this article is available online.

### Acknowledgements

S.J.D is a Jenner Investigator, Lister Institute Research Prize Fellow and Wellcome Trust Senior Fellow (106917/Z/15/Z). K. Matuschewski was supported by the Max Planck Society and grants from the European Commission (EviMalaR Network of Excellence #34) and the Chica and Heinz Schaller Foundation. O.S. was funded in part by the Laboratoire d'Excellence ParaFrap (ANR-11-LABX-0024). J.C.R.H. was funded by grants from The Royal Society (University Research Fellowship UF0762736/UF120026 and Project Grant RG130034) and the National Centre for the Replacement, Refinement & Reduction of Animals in Research (Project Grant NC/L000601/1). The funders had no role in study design, data collection and analysis, decision to publish or preparation of the manuscript. Parts of the illustrations used in the synopsis image were obtained from Servier Medical Art, (https://smart.servier.com) licensed under a Creative Commons Attribution 3.0 Unported License.

### Author contributions

Experiment design: KMa, OS and JCRH; Transgenic parasites CSP[SIINFEKL] and UIS4[SIINFEKL] : OS; Experiments and data analysis: KMü, MPG, MR, OS and JCRH; Adenovirus AdOVA: AR-S, AVSH and SJD; Manuscript writing: MPG and JCRH; Manuscript comments and revision: All authors.

### Conflict of interest

A.R.-S., A.V.S.H. and S.J.D. are named inventors on patent applications relating to malaria vaccines, adenovirus vaccines and immunisation regimens.

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
