## [Review Process File · EMBO Molecular Medicine]

Low immunogenicity of malaria pre-erythrocytic stages can be overcome by vaccination

Katja Mueller, Matthew Gibbins, Mark Roberts, Arturo Reyes-Sandoval, Adrian Hill, Simon Draper, Kai Matuschewski, Olivier Silvie, and Julius Hafalla

DOI: [10.15252/emmm.202013390](https://doi.org/10.15252/emmm.202013390)

Corresponding author: Julius Hafalla (julius.hafalla@lshtm.ac.uk)

Review Timeline:

Transfer from Reviews Common:	3rd Sep 20
Editorial Correspondence:	9th Sep 20
Editorial Decision:	1st Oct 20
Revision Received:	26th Nov 20
Editorial Decision:	24th Dec 20
Revision Received:	3rd Feb 21
Accepted:	5th Feb 21

Editor: Zeljko Durdevic

Transaction Report:

Review
COMMONS

This manuscript was transferred to EMBO Molecular Medicine following peer review at Review Commons.

Review #1

1. How much time do you estimate the authors will need to complete the suggested revisions:

Estimated time to Complete Revisions (Required)

(Decision Recommendation)

Between 1 and 3 months

2. Evidence, reproducibility and clarity:

Evidence, reproducibility and clarity (Required)

In this study by Müller et al, the authors study if immunogenicity is an adequate predictor for vaccine development in malaria and more precisely against malaria pre-erythrocytic stage. For that they used two different strains of the murine parasite *Plasmodium berghei*. They based their study on the use of the MCH I restricted epitope SIINFEKL to follow CD8 T cell responses. For that, they integrated the sequence SIINFEKL sequence into the protein CSP expressed by the infective form sporozoite and at the end of the sequence of the protein UIS4 expressed exclusively by the exo-erythrocytic forms (EEF) of the parasite. They compared then the CD8+ T cell responses elicited by each strain of the parasite and came to the conclusion that whilst antigen origin results in very different immunogenicity responses both sporozoite and EEF expressed antigens elicit antigen-specific effector CD8+ T cell responses with a high level of protection. ****Major comments:**** Whilst rationale of parasite strain design is adequate and well-performed and the concept of low immunogenicity novel potentially interesting, there are several methodological flaws that make the conclusions somewhat speculative and need to be addressed to really support the conclusions. Given the fact that authors are top-level scientists in the malaria vaccinology field, I am confident that they can address the following comments that will help to improve the manuscript and its impact; -General comment: this reviewer was expecting a little more of deep analysis of immune responses elicited by the transgenic parasites that authors developed and not only a superficial analysis, how about TRM cells, what is the endogenous responses to SIINFEKL without transferring CD8 + T cells from OTI mice? This should be addressed -Fig 2-3: Authors compared the CD8+ T cell responses elicited by the two different strains of *P.berghei*. In order to evaluate if the two strains allowed to track anti-SIINFEKL, they immunized mice with both irradiated parasite strains or their control WT. To track these responses mice were adoptively transferred with CD8+ T cells from OT-I mice and immunized with irradiated parasites. They track responses by using a SIINFEKL tetramer expressing CD8+ T cells in the blood and the marker for antigen-experienced T cells CD11a. The problem here is that with the strategy of gating on Fig2a, it is not clear if they want to track the responses from adoptively transferred CD8+ T cells to vaccine or the endogenous CD8+ T cell responses. In any case, the results are potentially interesting but need clarification. Fig 2: only the responses on

the spleen are studied. In order to support the statement about the two different kinds of immunization, they should assess the responses on the liver. There is also a lack of methodology in flow cytometry analysis, a viability stain is not used, the gating is not determined by FMO (fluorescence minus one) controls and seems aleatory. For activation markers in order to assess the impact of the vaccination authors have to use gating that is already established by some of the papers they mentioned (i.e: Harty lab's studies), it is difficult to evaluate the responses if we don't know how many of the CD11a/Cd49d cells are Memory effector or effector (CD62L and CD44 markers). Moreover, the CD11a label should be CD11ahi and is not stated anywhere. Line 165: the statement "massive proliferative activity" is not supported by the figure, moreover there are numbers to support the statement. IFNg and other cytokines production seems too low and the stimulation assay is poorly performed because CD8 were restimulated ex-vivo only with SIINFEKL peptide in the absence of APC (antigen-presenting cells) with Brefeldin A. Also Authors omitted negative controls (without SIINFEKL Brefeldin A) to be certain that IFNg production is due to SIINFEKL. Again we don't if they are OTI or endogenous cells. -Fig5. Are the cells from Fig5a,b SIINFEKL positive cells or only CD11a and IFNg? Are they OTI? Controls are missing to show a real IFNg production due to the ex vivo stimulation. -Fig 6. no percentages are shown in the cytometry plots, figure 6d and c seem to be inverted. An interesting observation is that the level of protection against both strains of parasites is the same when vaccinated mice with AdOVA are challenged. The authors make the interpretation that immunogenicity does not predict effector responses. This is one of the central conclusions of the paper. The authors only show level of protection but don't characterize the phenotype of CD8+ T cells in the liver of vaccinated and challenged mice. Can cells from Fig6a be found in the liver? Are they liver TRm (resident memory CD8+ T cells), known to be an important class of cells for protection against malaria. **Minor comments:** -For the two strains, authors should show the patency in comparison with WT parasites (currently presented as data not shown) -Gating strategy for markers is missing, FMO as well -Fig 6: how did the authors measure Sterile protection and Relative parasite load?

3. Significance:

Significance (Required)

The present study could provide important insight in the field of malaria vaccinology. By using cutting edge molecular biology to express the MCHI restricted epitope SIINFEKL at different stages of the pre-erythrocytic stage of Plasmodium and used it as a surrogate marker to evaluate the CD8+T cell response to infection. The authors attempt to provide proof of the concept of vaccine design by evaluating if accessibility/immunogenicity of the antigen is a decisive feature on vaccine design. Nevertheless, the potential of this study demands to be placed in the context of precedent studies that defined pre-erythrocytic stage CD8+ T cell responses. The authors failed to fully exploit the tools that they developed (transgenic parasite) by overlooking the last studies describing the importance of CD8+ liver-resident memory CD8+ T cells from Health's laboratory or well characterized CD8 T cells responses defined by Harty's laboratory. If well placed in the context (after revisions) this study will not only be fundamental to the malaria field but to other infectious diseases as well.

Field of expertise: malaria immunology, vaccinology, immunomodulation, CD8+ T cell responses

Review #2

1. How much time do you estimate the authors will need to complete the suggested revisions:

Estimated time to Complete Revisions (Required)

(Decision Recommendation)

Between 1 and 3 months

2. Evidence, reproducibility and clarity:

Evidence, reproducibility and clarity (Required)

The manuscript by Mueller and Gibbins et al titled "Low immunogenicity of malaria pre-erythrocytic stages can be overcome by vaccination" compares how transgenic *P. berghei* parasites expressing SIINFEKL epitope from ovalbumin, as part of CSP or UIS4 present the respective epitope and how immune responses occur to each of the mutants, mostly in mice pre-treated with 2×10^6 OT-I cells expressing a SIINFEKL-specific TCR. Their data show that when in their normal location CSP is much better than UIS4 to elicit an immune response, and that increasing UIS4 (by raising irradiated parasite numbers) does not greatly improve to reduce the difference. Finally the authors show that mice immunized with ovalbumin can reduce liver infection of either CSP-SIINFEKL or UIS4-SIINFEKL sporozoite challenge infection. The experiments presented by the authors are in my view well done and controlled, but I feel that sometimes conclusions are a bit beyond what the experimental readouts allow for.

3. Significance:

Significance (Required)

In Fig 1 the authors show how mutants were made and that proteins with associated SIINFEKL to CSP or UIS4 localise to correct place. (could all be supplementary or Supplementary Figure 1c, d could be included in Fig 1). In Fig 2a is shown the gating of SIINFEKL-specific CD8+ T cells (could be supplementary). In Fig 2b the authors show that the highest CD8T cell specific for SIINFEKL is on the first day analysed (d4) and I would like to see how day 2 and 3 would look like. Specially because proliferative differences don't seem massive to me, CFSE should decrease with each cell division and reach different fluorescence values if replication numbers differ. However here the CFSE fluorescence signal is similar on d5 Fig 2c, indicating a similar number of replicative rounds, but probably a different starting numbers of cells that would replicate. Or that

CFS labelling was too low to allow distinguishing the number of replicative rounds occurring in that time. so when the authors conclude that proliferative activity was 6x larger than that observed with UIS4SIINF EKL sporozoites, i think they would have to show before that numbers of cells prior to replication was the same Figs 3 and 4 show that response to CSP is stronger than response to UIS4, and in the spleen larger than in the liver and that this was true for mice adoptively transferred with OT-I cells prior to intravenously immunisation or without that, and that with the transfer responses were much higher. Fig 5 show that increasing # of irradiated UIS4SIINF EKL 8x does not bring levels of response to anywhere close than the observed against 1x CSPSIINF EKL. and fig 6 show that if a response is obtained (in the case with an adenovirus expressing ovalbumin which will generate e a response recognising SIINF EKL) both CSPSIINF EKL or UIS4SIINF EKL infection challenge can be blocked and protective immunity equally achieved. I think it would be nice to show when is infection stopped in these two groups os mice, but looking at EEF in the liver if the two groups of mice. Also the authors could show that an adenovirus carrying UIS4 (and CSP) would result in the same as observed here with the ovalbumin one). I also think the authors should discuss the advantages and problems of the two SPZ and PVM locations, assuming that indeed an adenovirus carrying UIS4/CSP would also result in similar protection upon challenge, regarding potential boost from natural Infection, and how variable/conserved each of the proteins are and what could be expected in field trials ion the falciparum counterpart.

Review #3

1. How much time do you estimate the authors will need to complete the suggested revisions:

Estimated time to Complete Revisions (Required)

(Decision Recommendation)

Less than 1 month

2. Evidence, reproducibility and clarity:

Evidence, reproducibility and clarity (Required)

****Summary**** The study by Müller et al. uses the rodent malaria parasite Plasmodium berghei, which gives access to the mouse in vivo infection model. The authors investigate the initiation and development of CD8+ T cell responses during parasite liver-stage infection when the peptide antigen SIINF EKL is presented either early or late during liver-stage infection by fusing SIINF EKL to the Circumsporozoite Protein (CSP SIINF EKL) or the Up-regulated in Infective Sporozoites 4 protein (UIS4 SIINF EKL). CSP is expressed already in the motile sporozoite that invades the liver while UIS4, is expressed only in the later exoerthrocytic forms developing in the infected hepatocytes. Using the SIINF EKL peptide the authors can control for epitope differences between CSP

and UIS4 and it allows them to use the OT-I transgenic mouse line that produces high numbers of CD8+ T-cells that specifically recognizes MHC class I presented ovalbumin (OT-I cells), which can be used in adoptive transfer experiments. Using this approach the authors provide detailed kinetic and phenotypic analysis of CD8+ T-cell responses to CSP SIINF EK and UIS4 SIINF EK antigen-specific response in vivo and ex vivo, chiefly using FACS. The authors found that despite having different timing of expression and immunogenicity both CSP SIINF EK and UIS4 SIINF EK induce similar levels of CD8+ mediated targeting, resulting in a high degree of sterile immunity in a vaccination challenge experiment. ****Major comments**** Overall the manuscript is well written, concise and with a clear narrative. The conclusions drawn from the study are well supported by the data presented, and the experiments are thoroughly controlled and sufficiently replicated. The manuscript is suited for publication but presentation of key concepts and data could be made clearer and impact enhanced if the authors address some of the following commentary. 1.Introduction -The authors assume a reader familiarity with the use of ovalbumin, the SIINF EK epitope, the transgenic T-cell receptor OT-1 mice and adoptive transfer experiments to assay immunogenicity. These concepts are not comprehensively introduced in the introduction, and the relationship between these tools are not delineated sufficiently to allow the non-expert reader to follow the logic and methodology of the experiments right from the start. Background information given in Results (Line 139-144, 204-208) and Discussion section, (Line 288-293) could with advantage be synthesized into one paragraph and presented in the introduction to bring all readers onboard from the start. 2.Results Page 11 Line 255-260, Figure legend Fig. 6 page 27 Line 258-665 The punchline of the paper is that despite differences in immunogenicity between γ -irradiated CSP SIINF EK or UIS4 SIINF EK sporozoites, both CSP SIINF EK or UIS4 SIINF EK are targets of protective CD8+ T-cell responses resulting in sterile immunity in a challenge following vaccination with full-length ovalbumin in OT-I cell recipient mice. This section is a cornerstone for the conclusions of the paper and would benefit from being better supported by its explanatory text and presentation of data. Firstly, there is a mix up, between panel 6C and 6D, where 6C shows "% Sterile protection" and 6D shows "Parasite load in the liver", while it says the opposite in main text and figure legend. Secondly, while it is clear that qPCR is used to measure liver parasite load at 24 hours after challenge. It is not immediately clear from neither main text nor figure legend that sterile immunity is measured by microscopy on blood films. The use of the term "sterile immunity" naturally implies this to the initiated reader, but it should be spelled-out that this was the case and that it was monitored from day 3-14 following challenge, which is outlined only in the methods section. Rewriting and restructuring this section to make this clearer would greatly help guide the reader through the results. Currently it reads at first pass as if qPCR on liver samples harvested at 42 hours was used to generate the data in both 6C and 6D. Thirdly, protective efficacy here is given as a percentage of those mice that become protected, presumably remaining negative by day 14. Authors should provide the actual blood stage parasitaemia in graph or table format in Figure 6 or as a supplemental figure to show that sterile immunity is obtained and maintained until day 14, and that in the control groups patency develops as normal. This will also give clearer insight into how many mice developed patency in the control groups and at what point break-through was observed. In a similar vein, qualitative and / or quantitative presentation of microscopy data of EEFs (as presented in Figure 1C) would strengthen conclusions drawn from the qPCR parasite liver load data. Fourthly, the authors should also comment on why there is such a great variation in the number of

mice used in the different studied groups, it says maximum of n=11 mice per group but one group only has as n=3 mice and another n=4, and make a convincing argument this does not affect the conclusions drawn and statistical analysis undertaken. Finally, the authors characterise CD8+ T-cell responses in absence of preceding OT-I adoptive transfer but do not report on whether the ovalbumin-immunization was tried on mice without preceding OT-I cell transplant. Was this tried? If not authors should discuss whether this is likely to be successful or not for readers to understand if both sporozoite and EEF presented antigens are likely to induce sterile immunity in a natural setting without artificial enrichment for epitope specific T-cells. 3. Line 723, 725 clarify if data is from independent biological repeats, i.e. different infected mice fed to different pots of mosquitoes, in which case the data is sufficiently replicated. **Minor comments:** 1- Page 6 Line 132 Please show blood-stage infection data / growth rates for CSP SINFEKL and UIS4 SINFEKL compared to WT as supplemental figure, if available. 2- Figure 1. If quantitative data is available for EEF, as indicated by the mean numbers with SD given within the microscopy pictures it would be nice to see these plotted. Does the reduced EEF numbers for CSP SINFEKL compared to UIS4 SINFEKL and WT mean anything? If not perhaps worth stating this in figure legend, or consider different presentation. Distracting when looking at the figure. 3- Figure 2. Would benefit from a panel with a simple schematic that shows the overall experimental design with irradiation of sporozoites, OT-1 transfer, administration of parasites and sampling with the timings for each event clearly marked out. 4- Figure Panel 2b would benefit from the in-figure legend stating CSP SINFEKL + OT-1, UIS4 SINFEKL + OT-1, WT + OT-1 and OT-1 only. Similar to as in Figure 3. 5- Figure 3 and Figure 4. Label within figure more clearly what is being measured, i.e. what is the difference between panels a,b,c vs. d,e,f (e.g. intravenously v.s. intradermal administrations), gets confusing since Figure 3 and Figure 4 are very similar within the figures (a,b,c vs. d,e,f) and between the figures. 6- Figure 3, 4, The Panel indicating letters (a, b, c, d...) become smaller as figures get bigger and become hard to read for Figure 3 and Figure 4. 7- Line 158 - Throughout manuscript, when it says administration of WT, CSP SINFEKL and / or UIS4 SINFEKL sporozoites it would be good to always have it preceded by irradiated when referring to irradiated sporozoites, e.g. Line 158 and only use sporozoites on its own when referring to live sporozoites (or even better spell out also when using live sporozoites e.g. Line 255). 8- The authors could measure the total amounts of IFN- γ being secreted in the tissues after immunization to both antigens and investigate if the level of other IFN- γ secreting cells might compensate for the weak response of CD8+ T cells, particularly against UIS4. If completed it has the potential to help the authors to in more detail understand the mechanism and contributing factors to the successful CD8+ T-cell targeting of UIS4. *Suggested extra experiments:* 9- As antigen protection is dependent not only cellular response but also on antibody responses induced against the antigens, authors should analyze by ELISA IgG and IgM responses induced against the two antigens 10- Page 6 Line 121-123 the authors reason that addition of the SINFEKL epitope to the immediate C-terminus of the UIS4 protein might confer enhanced antigen presentation through increase MHC-I antigen presentation. Supplementary experiments particularly a MHC I stabilization assay might help confirm this. Complementary experiments looking at MHC-II antigen presentation to APC would also be very relevant. 11- The authors have described the effect of both antigens in the response of CD8+ T cells and the generation of memory. A more detailed characterization of the differential phenotypes of memory in CD4 and CD8 T cells in the spleen and liver following immunostimulatory therapy would increase the relevance of the data presented.

3. Significance:

Significance (Required)

This paper would be of interest to malaria parasite biologists, immunologists and vaccinologists alike. The significance of this paper is three-fold. Firstly, the authors demonstrate contrasting immunogenic profiles between a sporozoite and EEF presented antigen. They comprehensively characterize respective CD8+ T-cells responses, with the sporozoite expressed antigen displaying enhanced immunogenicity compared to the the EEF expressed antigen. Secondly, the authors demonstrate that despite these stark differences in immunogenicity, both the sporozoite and EEF expressed antigens are effective targets of epitope specific CD8+ T-cell responses capable of eliciting sterile immunity. This has the important implication that low immunogenicity as defined by conventional immunological assay fails to capture all antigens that are capable of inducing sterile immunity, and thus could be prioritized as vaccine targets, but instead risks leading investigators down a path where so to speak "the baby is thrown out with the bath water". Thirdly, this work shows for the first time that EEF expressed antigens are potential vaccine targets, and thus effectively expands the pool of available pre-erythrocytic vaccine targets for the research community to explore. The data presented here should thereby lead to a complete revision in how pre-erythrocytic vaccine candidates are identified and prioritized. In terms of basic biology, the fact that CD8+ T-cells are critical in mediating immunity is been well established, however this paper greatly enhances our understanding of exactly how the dynamics, magnitude and quality of CD8+ T-cell responses are modulated by the timing of antigen expression. *Keywords for main reviewer expertise:* Malaria, Plasmodium berghei, genetic manipulation, host-parasite interactions *Keywords for ECR co-reviewer expertise:* Immunity, host-pathogen interactions.

Initial plan – Response to Reviewers' Comments

We thank the three reviewers for their comments, which on balance were very positive and supportive.

The fundamental relevance and translational impact of our manuscript were reinforced by R1 and R3: The present study "could provide important insight in the field of malaria vaccinology ..." (R1), "...lead to a complete revision in how pre-erythrocytic vaccine candidates are identified and prioritized ..." (R3) and "... greatly enhances our understanding of exactly how the dynamics, magnitude and quality of CD8+ T-cell responses are modulated by the timing of antigen expression ..." (R3).

R1 acknowledged our use of "using cutting edge molecular biology (techniques)" and our efforts to "... provide proof of the concept of vaccine design by evaluating if accessibility/immunogenicity of the antigen is a decisive feature on vaccine design ...". R3 emphasised that the "... manuscript is well written, concise and with a clear narrative..." and that "... conclusions drawn from the study are well supported by the data presented, and the experiments are thoroughly controlled and sufficiently replicated".

We have now revised the manuscript based on the reviewer's comments and clarified valid concerns. Together, we consider the review process very helpful to further enhance the impact of our study. At this point, we have also prepared a Graphical Abstract that summarises the key findings of the manuscript.

We address the Reviewer's Comments below:

Reviewer 1

1. General comments: ... a little more of deep analysis of immune responses elicited by the transgenic parasites ... how about TRM cells, what is the endogenous responses to SIINFKEL without transferring CD8 + T cells from OTI mice?

... the potential of this study demands to be placed in the context of precedent studies that defined pre-erythrocytic stage CD8+ T cell responses. ... the importance of CD8+ liver-resident memory CD8+ T cells from Heath's laboratory

We agree with the reviewer that the field of malaria pre-erythrocytic immunology is fast-moving. This is exemplified by the more recent identification of resident memory CD8+ T cells (T_{RM}) that patrol hepatocytes against pathogens, including malaria pre-erythrocytic stages by the Heath laboratory. Whilst the focus of our current work is on assessing the timing of expression and immunogenicity of pre-erythrocytic antigens as crucial features for vaccine design, we concur with the reviewer that the analysis of T_{RM} is of remarkable interest to the malaria field, which we believe is currently out of the scope of the current study. Nonetheless, we have now included this in the discussion to link the findings of our study with the evolving field of T_{RM} (**lines 409-420**).

Nevertheless, we characterised CD8+ T cells using techniques that are commonplace in studying immunological response in the malaria field, while also adapting our approach to probe responses using more physiological proxies.

Whilst we did not specifically phenotype for T_{RM} , we measured CD8+ T cell responses in the livers of immunised mice, utilising isolation methods for intrahepatic or liver-infiltrating lymphocytes (Goossens et al., 1990, PMID: 2202764). CD8+ T cells from the livers of mice immunised by irradiated sporozoites delivered intravenously were analysed following adoptive transfer of naïve CD8+ T cells (**Figure 3**), as well as quantifying endogenous CD8+ T cells (**Figure 4B-D**). We also assessed CD8+ T cells from the livers of mice immunised with irradiated sporozoites delivered intradermally (as a proxy for the natural route of infection and parenteral vaccine administration; **Figure 4E-G**).

As mentioned above, the response from endogenous CD8+ T cell, that is without adoptive transfer of naïve CD8+ T cells, are shown in **Figure 4A-C** (intravenous immunisation) and **Figure 4D-F** (intradermal immunisation).

We apologise to the reviewer if these results were not obvious. Accordingly, we have reconfigured the figures to i) colour-coordinate intravenous from intradermal administration of sporozoites in **Figure 4**, and ii) to allow differentiation of pentamer staining from IFN- γ production in **Figure 3**.

2. ... the strategy of gating on Fig2a, it is not clear if they want to track the responses from adoptively transferred CD8+ T cells to vaccine or the endogenous CD8+ T cell responses. In any case, the results is potentially interesting but need clarification.

The purpose of adoptively transferring naïve CD8+ T cells was to augment the frequencies of naïve precursors. We used K^b-SIINFEKL pentamers to visualise the developing CD8+ T cell response, which is a combination of both OT-I and endogenous responses. As mentioned above, we have compared the kinetics of the CD8+ T response in mice administered with OT-I cells (**Figure 3**) or endogenous responses in the absence of OT-I cells (**Figure 4**).

3. Fig 2: only the responses on the spleen are studied. In order to support the statement about the two different kinds of immunization, they should assess the responses on the liver.

As pointed out by the reviewer, responses in the liver were not performed for **Figure 2**, but were assessed in **Figures 3 to 5**.

4. ... lack of methodology in flow cytometry analysis, a viability stain is not used, the gating is not determined by FMO ... controls ... For activation markers in order to assess the impact of the vaccination authors have to use gating that is already established by some of the papers they mentioned (i.e: Harty lab's studies), ... the CD11a label should be CD11ahi and is not stated anywhere.

In our gating strategies we relied on the population of cells with a larger FSC value (healthy cells). In previous experiments we established this population to represent the desired population by staining cells with and without Live/Dead dye. Accordingly, we have found this first gating strategy to be satisfactory and consistently excluded dead cells and cell debris.

We apologise for not showing FMOs and have now included exemplary flow cytometry strategies in **Supplementary Figure 2 and 5** to illustrate how we gated for CD8+ T cells from blood, spleen and liver. In addition to FMOs we gated for markers in our pentamer and surface stain panel and restimulation cytokine panel.

We have focused on the enumeration of antigen-specific responses using K^bSIINFEKL pentamer staining, and the measurement of the effector molecule IFN- γ for the evaluation of responses to SIINFEKL. In both methodologies, responses were co-stained with CD8 and CD11a. The utilisation of the CD11a marker and identifying CD11a^{hi} populations in models of infections were established by the Harty and Badovinac laboratories (Rai et al., 2009, PMID: 19933864). CD11a^{hi} populations discriminate antigen-experienced but not inflammation-driven responses, particularly when analysing polyclonal populations of CD8+ T cells. We have referenced the original publication in the manuscript. Moreover, we have corrected mentions and labels of CD11a+ to CD11a^{hi} throughout the manuscript. Thus, in addition to K^bSIINFEKL pentamer and IFN- γ stainings, the CD11a marker was used as a confirmatory marker for antigen-driven activation. Furthermore, we also stained the cells for canonical activation markers: CD49d, CD62L and CD44. It is notable that the numbers of CD11a^{hi}, CD49d^{hi}, CD62L^{lo}, CD44^{hi} co-stain with K^b-SIINFEKL pentamer (**Figure 2 and Supplementary Figure 3**), indicating that the identified cells are of effector/ effector memory phenotypes.

5. Line 165: the statement "massive proliferative activity" is not supported by the figure, moreover there are numbers to support the statement.

We have toned down the term from "massive" to "greater". We have also altered the sentence to read "... immunisation with CSP^{SIINFEKL} sporozoites led to greater expansion of Kb-SIINFEKL+ CD8+ T cells, 6x larger than that observed with UIS4^{SIINFEKL} sporozoites..." (**line 195-196**), which is in agreement with that shown in **Figure 2D, E**.

6. IFN γ and other cytokines production seems too low and the stimulation assay is poorly performed because CD8 were restimulated ex-vivo only with SIINFEKL peptide in the absence of APC (antigen-presenting cells) with Brefeldin A. Also Authors omitted negative controls (without SIINFEKL Brefeldin A) to be certain that IFN γ production is due to SIINFEKL. Again we don't if they are OTI or endogenous cells.

We have utilised stimulation and flow cytometry protocols that are widely used in the malaria pre-erythrocytic stage field (Hafalla et al., 2013, PMID: 23675294; Jagannathan et al., 2015, PMID: 25520427), as well as other fields (Hosking et al., 2014, PMID: 25015828, Nakiboneka et al., 2019, PMID: 30459072). Notably, CD8⁺ T cell responses to this eukaryotic pathogen have been widely published to be much lower, in contrast to those evoked by viral and bacterial pathogens (Schmidt et al., 2008, PMID: 18780790).

As suggested, we have now included the corresponding negative controls (restimulation without peptide) in a new **Supplementary Figure 6**.

7. Fig5. Are the cells from Fig5a,b SIINFEKL positive cells or only CD11a and IFN γ ? Are they OTI? Controls are missing to show a real IFN γ production due to the ex vivo stimulation.

The responses shown in **Figure 5** were stimulated with SIINFEKL and stained for CD8⁺ (gated), CD11a^{hi}, IFN- γ ⁺. The mice did not receive OT-I cells, thus the data reflects the endogenous response.

8. Fig 6. no percentages are shown in the cytometry plots, figure 6d and c seem to be inverted.

We have now included the percentage in the flow cytometry plots. We apologise for the inversion of **Figures 6C and D**; this has now been corrected to match the figure legend.

9. For the two strains, authors should show the patency in comparison with WT parasites (currently presented as data not shown)

We have now included the patencies of WT, CSP^{SIINFEKL} and UIS4^{SIINFEKL} parasites in **Supplementary Figure 1f**. The transgenic parasites exhibit similar patencies to WT parasites.

10. Fig 6: how did the authors measure Sterile protection and Relative parasite load?

We have detailed the measurements of sterile protection and relative parasite load (level) in the Methodology section. Both methodologies are standard procedures in the malaria pre-erythrocytic stage field.

Reviewer 2

11. In fig1 the authors show how mutants were made and that proteins with associated SIINFEKL to CSP or UIS4 localise to correct place. (could all be supplementary or Supplementary Figure 1c, d could be included in Fig1). In Fig2a is shown the gating of SIINFEKL-specific CD8⁺ T cells (could be supplementary).

We deem the depiction of CSP and UIS4 in **Figure 1B and C** to be important for the concept and impact of the study. We would like to adhere to common practice in immunology studies and keep one representative flow cytometry gating strategy in the main paper (in **Figure 2B**), to illustrate an example of our analysis methods going forward.

12. In Fig 2b the authors show that the highest CD8T cell specific for SIINFEKL is on the first day analysed (d4) and I would like to see how day 2 and 3 would look like. specially because proliferative differences don't seem massive to me, CFSE should decrease with each cell division and reach different fluorescence values if replication numbers differ.

however here the CFSE fluorescence signal is similar on d5 fig 2c, indicating a similar number of replicative rounds, but probably a different starting numbers of cells that would replicate. Or that CFS labelling was too low to allow distinguishing the number of replicative rounds occurring in that time.

so when the authors conclude that proliferative activity was 6x larger than that observed with UIS4SIINFEKL sporozoites, i think they would have to show before that numbers of cells prior to replication was the same

This is a good suggestion, but unfortunately, we did not perform CFSE experiments on days 2 and 3. We agree that the resulting CD8+ T cell responses to both parasites seem to have similar replication rounds (number of cell division), yet the frequencies of those recruited to the immune response are much more elevated in the CSP^{SIINFEKL} as compared UIS4^{SIINFEKL} parasites (5.05 vs 0.84, respectively – as shown in **Figure 2D**). A better representation is shown in **Figure 2E**, which is gated on Kb^{SIINFEKL}₊, CD11a^{hi}, CD8+ T cells.

For our study, we have used published and standard CFSE labelling protocols (Lundie et al., 2008, PMID: 18799734).

In light of Reviewer 1 and 2 both commenting on our use of terminology regarding proliferation, we altered and corrected the text in the manuscript to address that there is a 6x increase (5.05 vs.0.84) in recruitment of SIINFEKL-specific CD8+ T cells rather than proliferation (**line 195-196**). The same number of cell divisions were undergone, however the level of expansion was greatly increased when mice were immunised with CSP^{SIINFEKL}.

13. I think it would be nice to show when is infection stopped in these two groups os mice, but looking at EEF in the liver if the two groups of mice.

We believe that the reviewer is referring to the outcomes of the protection experiments. We utilised a widely used quantitative PCR method to quantify the EEF in the liver after challenge of vaccinated mice. We agree with the reviewer that it will be interesting to determine whether the kinetics of killing by vaccine-induced CD8+ T cells of CSP^{SIINFEKL} and UIS4^{SIINFEKL} parasites are different. While presently out of the scope, we would have to establish *ex vivo* quantitative imaging and hope to advance on this in the future.

14. ... could show that an adenovirus carrying UIS4 (and CSP) would result in the same as observed here with the ovalbumin one).

The vaccine efficacy of an Adenovirus vaccine expressing CSP has been established (Rodrigues et al., 1997, PMID: 9013969; Bruña-Romero et al., 2001, PMID: 11553779; Gilbert et al., 2002, PMID: 11803063. Since there are no known 'immunodominant' CD8+ T cell epitopes in UIS4 such a vaccine construct is likely to only serve as negative control.. We and others have previously systematically screened for CD8+ T cell epitopes in the pre-erythrocytic stages, including from UIS4, of *Pb*, but experimental testing yielded only few peptides, with none from UIS4.

15 ... discuss the advantages and problems of the two SPZ and PVM locations, assuming that indeed an adenovirus carrying UIS4/CSP would also result in similar protection upon challenge, regarding potential boost from natural infection, and how variable/conserved each of the proteins are and what could be expected in field trials ion the falciparum counterpart.

We have now included these points in the discussion. Thus far, the consensus in the field is that T cell responses to pre-erythrocytic stage antigens are low in endemic areas (Heide et al., 2019, PMID: 30949162), and there is a striking paucity of data on the impact of boosting (primary infection vs. multiple infections) in the field (Doolan et al., 1993, PMID: 7680226); Khusmith et al., 1999, PMID: 10774643). Previous work in rodent models has demonstrated that boosting of T cell responses to liver stage antigens is poor (Murphy et al., 2013, PMID: 23530242), and this was also documented for CSP (Hafalla et al., 2003, PMID: 12847268). With the very low responses to UIS4^{SIINFEKL}, we

reasoned whether they could be enhanced by increased dose of immunisation. However, **Figure 5** rejected this hypothesis.

It is noteworthy that we selected CSP and UIS4 as the best characterized representatives of sporozoite and EEF vacuolar antigens, respectively. Following up on the reviewer's comments, it would be interesting to contrast the allelic diversity of sporozoite and EEF antigens, since this information will be important for vaccine design.

Reviewer 3

16. The authors assume a reader familiarity with the use of ovalbumin, the SIINFEKL epitope, the transgenic T-cell receptor OT-1 mice and adoptive transfer experiments to assay immunogenicity. These concepts are not comprehensively introduced in the introduction, and the relationship between these tools are not delineated sufficiently to allow the non-expert reader to follow the logic and methodology of the experiments right from the start. Background information given in Results (Line 139-144, 204-208) and Discussion section, (Line 288-293) could with advantage be synthesized into one paragraph and presented in the introduction to bring all readers onboard from the start.

We thank the reviewer for this important point and have now addressed this in the introduction which reads as follows:

"To control for epitope specificity, we generated Pb transgenic parasites that incorporate the MHC class I H-2-Kb epitope SIINFEKL, from ovalbumin, in either the CSP or UIS4 protein. The resulting transgenic parasites develop normally as wild-type (WT) Pb in the mosquito vector and mammalian host. However, SIINFEKL would be expressed at the same time and space as its respective Plasmodium protein, enabling the CD8+ T cell response against these proteins to be tracked in an epitope-specific physiological manner. In line with previous studies (8,15), to augment low numbers of CD8+ T cell in the naïve response, cells from OT-1 mice, which express SIINFEKL-specific TCRs on their CD8+ T cells, were initially adoptively transferred to mice prior to them receiving sporozoite immunisations" (lines 117-127).

17. Results Page 11 Line 255-260, Figure legend Fig. 6 page 27 Line 258-665 The punchline of the paper is that despite differences in immunogenicity between γ -irradiated CSP SIINFEKL or UIS4 SIINFEKL sporozoites, both CSP SIINFEKL or UIS4 SIINFEKL are targets of protective CD8+ T-cell responses resulting in sterile immunity in a challenge following vaccination with full-length ovalbumin in OT-1 cell recipient mice. This section is a cornerstone for the conclusions of the paper and would benefit from being better supported by its explanatory text and presentation of data.

Firstly, there is a mix up, between panel 6c and 6d, where 6c shows "% Sterile protection" and 6d shows "Parasite load in the liver", while it says the opposite in main text and figure legend.

We apologise for this error, which has now been corrected. We also added more explanatory text in the results section to avoid reader's missing the punchline and impact of our study, which reads as follows: "Strikingly, contrary to the differential CD8+ T cell responses induced by CSP and UIS4, there was no statistical difference in the protection observed when vaccinated mice were challenged with either CSP^{SIINFEKL} or UIS4^{SIINFEKL} sporozoites. Consistent with these findings, both groups of vaccinated mice challenged with either CSP^{SIINFEKL} or UIS4^{SIINFEKL} sporozoites exhibited sterile protection of comparable levels..." (lines 305-310).

18. Secondly, while it is clear that qPCR is used to measure liver parasite load at 24 hours after challenge. It is not immediately clear from neither main text nor figure legend that sterile immunity is measured by microscopy on blood films. The use of the term "sterile immunity" naturally implies this to the initiated reader, but it should be spelled-out that this was the case and that it was monitored from day 3-14 following challenge, which is outlined only in the methods section. Rewriting and restructuring this section to make this clearer would greatly help guide the reader through the results. Currently it reads at first pass as if qPCR on liver samples harvested at 42 hours was used to generate the data in both 6C and 6D.

We agree that this important point should be consistently described and have now added the necessary clarification in the results (**line 310-311**) and figure legend (**line 799-800**) to indicate that we used microscopy to assess blood smears for parasitaemia.

19: Thirdly, protective efficacy here is given as a percentage of those mice that become protected, presumably remaining negative by day 14. Authors should provide the actual blood stage parasitaemia in graph or table format in Figure 6 or as a supplemental figure to show that sterile immunity is obtained and maintained until day 14, and that in the control groups patency develops as normal. This will also give clearer insight into how many mice developed patency in the control groups and at what point break-through was observed.

We have now included prepatency in our manuscript to illustrate if and when non-vaccinated and vaccinated animals became parasitaemic. Mice were monitored up to day 14, after which they were deemed sterilely protected. This is found in the new **Supplementary Table 2**.

20. In a similar vein, qualitative and / or quantitative presentation of microscopy data of EEFs (as presented in Figure 1C) would strengthen conclusions drawn from the qPCR parasite liver load data.

We have included a graph detailing quantitative data of EEF counts as **Supplementary Figure 1E**.

21. Fourthly, the authors should also comment on why there is such a great variation in the number of mice used in the different studied groups, it says maximum of n=11 mice per group but one group only has as n=3 mice and another n=4, and make a convincing argument this does not affect the conclusions drawn and statistical analysis undertaken.

In this experiment (**Figure 6D**), we placed particular emphasis on the quantification of the liver load in AdOVA-immunized mice challenged with UIS4^{SIINFEKL} sporozoites. We included cumulative data from multiple challenge experiments. The other groups of mice serve as controls and consistently displayed high parasite loads in non-immunized or WT sporozoite-challenged controls and very low parasite loads in CSP^{SIINFEKL} sporozoite-challenged mice, respectively.

22. Finally, the authors characterise CD8+ T-cell responses in absence of preceding OT-I adoptive transfer but do not report on whether the ovalbumin-immunization was tried on mice without preceding OT-I cell transplant. Was this tried? If not authors should discuss whether this is likely to be successful or not for readers to understand if both sporozoite and EEF presented antigens are likely to induce sterile immunity in a natural setting without artificial enrichment for epitope specific T-cells.

We thank the reviewer for highlighting this point. We did not vaccinate mice in the absence of OT-I cells. Previous work with *Py* and *PbCSP*-based adenovirus vaccines yielded only up to 40% sterile immunity, despite up to 97% reduction in parasite load in the liver after challenge with viable sporozoites (Rodrigues et al, 1997, PMID: 9013969; Rodrigues et al., 1998, PMID: 9795385). Thus, we augmented the numbers of naïve antigen-specific CD8+ T cell precursors by adoptively transferring OT-I prior to vaccinating with recombinant adenovirus. This methodology was chosen in order to attain optimal levels of vaccine-induced *effector* CD8+ T cells producing IFN- γ in a single vaccination, and to obtain reliable frequencies comparable to those achieved by prime-boost vaccinations with recombinant adeno- followed vaccinia viruses, or with peptide-loaded dendritic cells followed by recombinant *Listeria*. Previous work by colleagues and ourselves have shown that in order to achieve sterile protection in both the *Py*- and *Pb*-Balb/c model, vaccine-induced CSP-specific CD8+ T cells must exceed a threshold of >1% of all CD8+ T cells in peripheral blood (Bruña-Romero et al., 2001, PMID: 11553779; González-Aseguinolaza et al., 2003, PMID: 14557672; Schmidt et al, 2011, 21460205). Moreover, B10 backgrounds (including C57BL/6) further increases the threshold necessary for sterile protection through a CD8+ T cell-extrinsic mechanism. In our current study, the mean frequencies of antigen-specific CD8+ T cells induced following adenovirus vaccination was

7.5% (**Figure 6C**), which translated to 80% sterile protection (combined data from CS^{SIINFEKL} and UIS4^{SIINFEKL} groups).

In the current manuscript, we believe that we have successfully provided proof-of-concept evidence to assess the timing of expression and immunogenicity of pre-erythrocytic antigens as crucial parameters for vaccine design. Nonetheless, we have added a comment in the discussion on our chosen approach to test for vaccine efficacy, and on the importance of achieving relatively high levels of CD8+ T cells to enable high vaccine efficacy:

“Regardless of their differing immunogenicities in the context of parasitic infection, we further demonstrated that both sporozoite and EEF antigens are effectively targeted by antigen-specific effector CD8+ T cells, which were generated by vaccination using priming and boosting with recombinant viruses expressing the epitope. This method of prime-boost using recombinant viruses has been consistently shown to induce high numbers of antigen-specific CD8+ T cells (39-43) necessary for protection(20). Importantly, mice harbouring similarly high levels of vaccine-induced, antigen-specific CD8+ T cells were comparably protected when challenged with either CSP^{SIINFEKL} or UIS4^{SIINFEKL} (lines 371-379).

23. Line 723, 725 clarify if data is from independent biological repeats, i.e. different infected mice fed to different pots of mosquitoes, in which case the data is sufficiently replicated.

The mosquito infectivity is from 14 different mosquito feedings and the sporozoite numbers per mosquito were calculated from 18 (UIS4^{SIINFEKL} and WT) and 21 (CSP^{SIINFEKL} n=21) independent infections.

24. Page 6 Line 132 Please show blood-stage infection data / growth rates for CSP SINFEKL and UIS4 SINFEKL compared to WT as supplemental figure, if available.

We have now included prepatency data for the two transgenic parasites (**Supplementary Figure 1f**).

25. Figure 1. If quantitative data is available for EEF, as indicated by the mean numbers with SD given within the microscopy pictures it would be nice to see these plotted. Does the reduced EEF numbers for CSP SINFEKL compared to UIS 4SINFEKL and WT mean anything? If not perhaps worth stating this in figure legend, or consider different presentation. Distracting when looking at the figure.

We have generated a graph depicting the numbers of EEFs developing *in vitro* from sporozoite of Huh7 cells from two independent experiments. This is now found in **Supplementary Figure 1E**.

26. Figure 2. Would benefit from a panel with a simple schematic that shows the overall experimental design with irradiation of sporozoites, OT-1 transfer, administration of parasites and sampling with the timings for each event clearly marked out.

We thank Reviewer 3 for this suggestion and have now included timelines of our experimental design for **Figure 2**, as well **Figures 3-6**.

27. Figure Panel 2b would benefit from the in-figure legend stating CSP SINFEKL + OT-1, UIS4 SINFEKL + OT-1, WT + OT-1 and OT-1 only. Similar to as in Figure 3.

We thank the reviewer for this suggestion. Since OT-1 transfer was done in all groups of mice, and hence, is not a distinctive feature, we have instead included a timeline on top of the graph with clear colour coding showing administration of OT-I cells prior to sporozoite immunisation (**Figure 2A**). We believe this is sufficient to guide the reader through the figure. Similarly, we reduced the labelling in **Figure 3**, and instead added a timeline as a reference for the experimental design (**Figure 3A**).

28. Figure 3 and Figure 4. Label within figure more clearly what is being measured, i.e. what is the difference between panels a,b,c vs. d,e,f (e.g. intravenously v.s. intradermal

administrations), gets confusing since Figure 3 and Figure 4 are very similar within the figures (a,b,c vs. d,e,f) and between the figures.

We thank the reviewer for this suggestion and have now added colour coding to **Figures 3 and 4** and an experimental schematic to guide the reader through the data. We have included segregation lines above the flow cytometry plots to further guide the reader, i.e. **Figure 3B-D** denotes Kb-SIINFEKL pentamer data, while **Figure 3E-G** denotes IFN- γ production following restimulation. Further, in Figure 4 segregation lines have been added and labelled to allow easy discernibility of **panels 4B-D** (intravenous immunisation of sporozoites) *vis-a-vis* **panels 4E-G** (intradermal immunisation of sporozoites).

29. Figure 3, 4, The Panel indicating letters (a, b, c, d...) become smaller as figures get bigger and become hard to read for Figure 3 and Figure 4.

This has now been adjusted, as suggested.

30. Line 158 - Throughout manuscript, when it says administration of WT, CSP SINFEKL and / or UIS4 SINFEKL sporozoites it would be good to always have it preceded by irradiated when referring to irradiated sporozoites, e.g. Line 158 and only use sporozoites on its own when referring to live sporozoites (or even better spell out also when using live sporozoites e.g. Line 255).

We have followed the advice of the reviewer including " γ -radiation attenuated" or "live" as appropriate (lines 188, 202-203, 216, 236, 248-249, 260-261, 280 and 299).

31. The authors could measure the total amounts of IFN- γ being secreted in the tissues after immunization to both antigens and investigate if the level of other IFN- γ secreting cells might compensate for the weak response of CD8+ T cells, particularly against UIS4. If completed it has the potential to help the authors to in more detail understand the mechanism and contributing factors to the successful CD8+ T-cell targeting of UIS4.

As antigen protection is dependent not only cellular response but also on antibody responses induced against the antigens, authors should analyze by ELISA IgG and IgM responses induced against the two antigens.

We thank the reviewer for raising interest on possible future directions for our study. We have specifically engineered SIINFEKL to be a part of either CSP or UIS4 and utilised an OVA-expressing adenovirus to focus on CD8+ T cell responses. However, we agree with the great idea from the reviewer that justifies further work in dissecting the multifaceted mechanisms underlying CD8+ T cell-mediated protection to malaria pre-erythrocytic stages, as well as future combinations to assess contributions of antibody responses.

32. Page 6 Line 121-123 the authors reason that addition of the SIINFEKL epitope to the immediate C-terminus of the UIS4 protein might confer enhanced antigen presentation through increase MHC-I antigen presentation. Supplementary experiments particularly a MHC I stabilization assay might help confirm this. Complementary experiments looking at MHC-II antigen presentation to APC would also be very relevant.

We have appended the SIINFEKL to the C-terminus of the UIS4, based on earlier studies in *Toxoplasma gondii* that the potency of an immunodominant epitope was associated with its C-terminal location, allowing for enhanced presentation by infected cells. Whilst this information is not defined for UIS4, studies on the basic biology of pre-erythrocytic stages have demonstrated for several ETRAMPs (UIS4 is a member of the ETRAMP protein family) that the C-terminus faces the host-cell cytoplasm, which might enhance exposure to the MHC I machinery. Our findings showing that vaccine-induced effector CD8+ T cell responses eliminate both transgenic parasites, argues against potential defects in antigen processing and presentation of SIINFEKL in both systems.

Again, we agree that future studies aiming at dissecting the molecular mechanisms of MHC-I antigen presentation in infected host cells and cross-presentation *via* MHC-II are warranted. Another long-standing goal of the community is to elude *Plasmodium* peptides from MHC molecules, similar to the pioneer work by Rammensee and co-workers. One potential, albeit challenging, research direction could be to focus on rare EEF-derived peptides, since they might prove to be excellent and hitherto neglected subunit vaccine candidates, as exemplified in the present proof-of-concept study.

33. The authors have described the effect of both antigens in the response of CD8+ T cells and the generation of memory. A more detailed characterization of the differential phenotypes of memory in CD4 and CD8 T cells in the spleen and liver following immunostimulatory therapy would increase the relevance of the data presented.

We entirely agree that a more detailed characterisation of the different phenotypes of not only memory CD8+, including T_{RM}, but also CD4+ T cell responses, is warranted. Whilst these suggestions clearly inspire further work using the transgenic parasites of this study by colleagues and ourselves, we believe that these are out of the scope of the current study.

9th Sep 2020

Dear Dr. Hafalla,

Manuscript Number: EMM-2020-13390

Title: Low immunogenicity of malaria pre-erythrocytic stages can be overcome by vaccination

Thank you for submitting your revised manuscript to EMBO Molecular Medicine. We will now re-evaluate your submission and will contact you again on completion.

1st Oct 2020

Dear Dr. Hafalla,

Thank you for the submission of your manuscript to EMBO Molecular Medicine. We have now heard back from the three referees who agreed to evaluate your manuscript. As you will see from the reports below, while referee #2 and #3 are overall supporting publication of your work, referee #1 highlights the interest of the study but also raises a concern regarding the provenance of the immune cells. We would strongly suggest that you take the time and perform experiments in congenic mice in order to address the referee #1 criticism in a major revision of the current manuscript to strengthen the main conclusions of the study.

Nevertheless, given the overall positive evaluation of the manuscript by the referees #2 and #3 we would be willing to consider your manuscript without an additional experimentation if you tone down the conclusions and discuss the limitations of the study in regard to the missing experiments with congenic mice. Addressing the reviewers' concerns in full, experimentally or in writing, will be necessary for consideration of your manuscript in our journal, and acceptance of the manuscript will entail a second round of review. Please note that EMBO Molecular Medicine encourages a single round of revision only and therefore, acceptance or rejection of the manuscript will depend on the completeness of your responses included in the next, final version of the manuscript. For this reason, and to save you from any frustrations in the end, I would strongly advise against returning an incomplete revision.

We would welcome the submission of a revised version within three to six months for further consideration. However, we realize that the current situation is exceptional on the account of the COVID-19/SARS-CoV-2 pandemic. Please let us know if you require longer to complete the revision.

I look forward to receiving your revised manuscript.

***** Reviewer's comments *****

Referee #1 (Remarks for Author):

This manuscript by Mueller et al studies if immunogenicity is an adequate predictor for vaccine development in malaria pre-erythrocytic stage. They used two strains of the murine parasite

P.berghei expressing the MCHI restricted epitope SIINFEKL at the sporozoite stage or at the exo-erythrocytic forms (EEF) of the parasite. I stand with my previous review concerning the Adequacy of model system, the novelty and the potential medical impact (design of malaria vaccine). I also appreciated the effort to address some of the comments and to increase the clarity of the manuscript especially with the schematics in the different figures.

Nevertheless, I observe that authors oversight some comments concerning technical issues. Technical procedures can appear to be negligible if the rationale is adequate (as is the case here) but good ideas have to be supported by adequate and outstanding technical procedures. If they don't have the data, I believe they can easily perform the experiments.

-Major comments:

Figure 2: authors transferred CD8 OTI cells in order to "augment low numbers of CD8+T cells" (line 124). I don't understand what is the point to augment low numbers of CD8+ T cells. Transfer of these cells is more to track the response rise against the SIINFEKL expression by the two different strains. This is absolutely adequate but in figure 2 authors cannot know if the cells that are tetramer positive are endogenous cells (from vaccinated mice) or the cells that they transfer. They don't use a congenic system used in reference 8 and 15 (CD45.1 OTI cells on CD45.2 recipient mice for example) that will allow to really track this response. This is even more important for the Figure 2E where they measure proliferation by CFSE dilution, thus non transferred cells would be also CFSE low/-.

Figure 3: Same problem than in figure 2. Tetramer positive cells can be also cells from immunized mice. I agree that the proportion is certainly small compared with the transferred cells but this is essential to support their arguments.

Figure 4: I would like to see the proportion of tetramer-positive cells here. I agree that cells are stimulated with peptide but a separate stain to support the specificity of the response is necessary.

Figure 5: same comments that on Figure 4 and more when authors are claiming that non-response against UIS4 SIINFEKL is observed or is low. They have to support that with the number of tetramer-positive cells.

Figure 6: same concern than above, responses from endogenous or OTI cells have to be determined.

Flow cytometry: Authors should be more careful with the utilization of FSC parameter to assess cell viability/health. The gold standard method to assess cell viability is using a dye. Moreover the gating on FSC is arbitrary. Authors said "that in previous experiments they established this population to represent the desired population staining cells with or without Live/Dead dye". Assuming that this strategy really works (change in the variability between experiments) I would like to see it.

It is not clear to me how many mice were used in each experiment, some figures state less than five mice. That has to be clarified.

Can authors rewrite their material and methods? Specially the part of lymphocytes isolation from livers (no Percoll gradient was apparently used to isolate lymphocytes or Did they use only liver cells as stated. In this case the gating using CD3 and CD8 cells is inappropriate, I would like to see a gating using another marker (CD45?) to define the erythrocytes population.

Referee #2 (Comments on Novelty/Model System for Author):

I think the model is the possible one for this type of studies, the medical impact is low just because this very far still from anything translational.

Referee #2 (Remarks for Author):

Mueller and Gibbins et al. in "Low immunogenicity of malaria pre-erythrocytic stages can be overcome by vaccination" show in an unbiased system that CSP, the main sporozoite antigen induces a higher immunological response than UIS4 a liver stage PV antigen, in a mouse model of liver stage malaria. The nice thing about their system (that inserts an OVA motif instead of a CSP one, or next to UIS4) is that allows to see how location of the same antigen influences the response.

pre-treating the animals with OT-I cells expressing a SIINFEKL-specific TCR specific, or quantifying only the newly generated one the authors show that mice produce more CD8T cell responding to SIINFEKLCSP than to SIINFEKLUIS4 .

Also the authors show increasing UIS4 immunisation levels does not compensate for lower immunogenicity, indicating that contribution of liver-stage natural infection should have null or minimal contribution for malaria immunity.

finally in a sledgehammer type of experiment, the authors show when 5-15% IFN+ CD11a+ specific CD8+T cells are generated following OVA adenoviral immunization, these cells can 19 days later abrogate infection of SIINFEKLCSP and SIINFEKLUIS4, showing that no matter how hidden the antigen is, a good vaccine with a strong response will be effective. this is however a very particular setting which says nothing beyond this exact particular case, as would be any of the other experimental settings the much less IFN+ CD11a+ specific CD8+T were induced. I recommend discussing appropriately

I believe Figure 2 (or a supplementary figure to add to it) needs quantification and statistics on the levels of CD11a , CD62L , CD49d and the proliferation of CFSE-labelled cells, as only the representative plot is shown.

in my opinion the title of the figures could be more consistent in form, sometimes being descriptive of the experiment, sometimes of the result, i would make it more uniform

I believe the authors should comment on how this work helps answers to the questions raised in the introduction ln. 95-97 and how their and other future answers will be key for antigen selection and design of future malaria vaccines

really very minor comments

I think ln. 50 needs a comma between of and and

in ln 216 "visualise" would probably be better replaced by identify or detect

ln 52 25 vs ln 271 35 years of livers age vaccines, although both are true, would probably be better to go for 35 in both instances

Referee #3 (Remarks for Author):

My initial review of the paper submitted by Müller et al. was that it is well written, with its conclusions and narrative being well supported by the data provided. This assessment stands and the manuscript has been further improved in a satisfactory manner. In my view this manuscript is now ready for publication, with some very minor revisions not requiring additional peer-review. The correction, rewrites, additional data and improvements to figures and the manuscript that was requested have been addressed adequately and are summarized below.

I had requested for some background information regarding the ovalbumin, the SIINFEKL epitope, the transgenic T-cell receptor OT-1 mice and adoptive transfer experiments to be made clearer up front in the introduction. This has been done and makes the manuscript instantly more accessible for interested readers unfamiliar with this particular system (Comment 16).

The take home message of the study is that despite significantly different immunogenicity profiles obtained when SIINFEKL is presented early (CSP-SIINFKL) compared to later (UIS4-SIINFKL) during the malaria liver stages, both SIINFEKL presentation strategies invoke sterile immunity. The text and figures (particularly figure 6) presenting these conclusions stood to benefit from some revisions to figure panels and their description in the main text, as well as a minor figure referencing mistake. This has been addressed (Comment 17-18).

Supplementary data showing prepatency as proportion of mice, but not actual parasitaemia, of immunized and control mice is now included as new Supplementary Table 2, showing the actual number of protected and infected mice of each group. This is acceptable and further strengthens the case of "equal protection" the data makes. There is however a minor problem with the sentence in the table legend "b brackets indicate the only a small proportion of mice...." (Comment 19).

I also requested qualitative and / or quantitative presentation of microscopy data of the EEFs assessed following immunization by qPCR in Figure 6D. It was perhaps clumsily worded on my behalf, but instead of data relating to Figure 6D, the data from Figure 1 C (WT, CSP-SIINFEKL and UIS4-SIINFEKL) was re-represented in a more quantitative manner in new Supplementary Figure 1E, which was also in fact what was requested in Comment 25, and thereby addresses this point instead.

It is even clearer from the new Figure S1E that CSP-SIINFEKL display lower EEF numbers, I asked to please comment if this is the case or state if not significant, I cannot find such a comment. Prepatency as measured by blood stage parasitaemia was also requested, and is now presented as new Supplementary Figure 1F (Comment 24). Convincingly and importantly the lower number of CSP-SIINFEKL EEFs does not appear to translate to a delay in pre-patency, and could be commented on in context of Figure S1E.

Would be good to now include similar confirmation of EEF numbers (based on microscopy) as a complement to qPCR data presented in Figure 6D if available, as requested initially, but not a prerequisite to publication. (Comment 20).

Comment 21, 23 regarding sample numbers and experimental repeats have been adequately addressed.

Figure 2 has now been fitted with an experimental timeline that greatly assists the reader in understanding the figure and experiments, and this has with benefit been extended to Figures 3-6 (Comment 26, 27).

Figure 3 and 4 are now much more clearly presented to the reader (Comment 28-29).

The clarification in text of when referring to live and irradiated attenuated or live sporozoites is a welcomed improvement (Comment 30).

Comment 31-33 suggest a number of experiments that could be used to explore, and likely reveal more of, the immunological mechanisms underpinning the disparity between measured CD8 T-cell responses to early and late presented liver stage antigens and their protective effect. Experiments of this nature would likely enhance understanding of the molecular mechanisms at play and could thereby potentially increase the impact of this paper, but are not essential to support the conclusions drawn in the paper in its current form, and are thereby not necessary for publication.

Similarly, exploring the effect of timing of antigen presentation (sporozoite and EEF presented) on not only CD8+ T-cell responses but also induction of sterile immunity in a more natural setting without OT-I adoptive transfer would strengthen this study and broaden the implications of the results. Nevertheless lack of thereof is in my view not a barrier to publication since it does not alter the proof-of-principle conclusions of this study regarding the timing of antigen expression, CD8 T-cell responses and sterile immunity. It has nevertheless been addressed in the revised discussion section by referral to previous published material on the subject of magnitude of CD8+ T-cell responses known to be associated with protection (Comment 22).

We thank the three reviewers for their evaluation, which continue to be very positive and supportive since the transfer from Review Commons. We also thank the *EMBO Molecular Medicine* editorial board for providing additional advice. We thank all the reviewers for appreciating our efforts in addressing the initial comments in order to enhance and improve the clarity of our manuscript. The addition of experimental timelines have been welcomed by the reviewers.

All reviewers reiterated the adequacy of the model systems used in our experiments. Both R1 and R3 further reinforced the novelty and the translational impact of the work.

We have further revised the manuscript based on the comments of the reviewers and advice of the editorial board. We have edited the abstract to fit the journal's requirement of 175 words. Please note that we have also updated the statistical analyses using Prism V9 (V7 was used in the first submission), resulting in some changes in the significance levels of a few figures; the statistical updates did not modify the conclusions of our work.

We address the comments from reviewers below:

1. **R1 further queried 'the point to augment low numbers of CD8+ T cells' (for Figures 2, 3 and 6 as opposed to Figures 4 and 5) and to 'know if the cells that are tetramer positive are endogenous cells (from vaccinated mice) or the cells that they transfer'. R1 also asked to use a congenic system to track the response to CSP^{SIINFEKL} and UIS4^{SIINFEKL}.**

In our manuscript, we have compared the kinetics of the CD8+ T cell response in mice that received OT-I cells (**Figures 2 and 3**) or did not (**Figure 4**); we refer the latter as the endogenous response to parasites. We decided only to perform pentamer staining for the former. This decision was based on a large body of published work from leading laboratories in the field (F Zavala (Johns Hopkins), R Nussensweig & V Nussenzweig (NYU Medical Center), M Nussenzweig (Rockefeller), A Hill (Jenner Institute, Oxford) and J Harty (Iowa), where endogenous responses (without transfer of transgenic CD8+ T cells of any parasite-derived specificity) are regarded to as very low in comparison to T cell responses to bacterial and viral infections, and that antigen-specific responses are reported as IFN- γ -producing CD8+ T cells, following stimulation with a cognate parasite-derived peptide. Pentamer or tetramer staining is performed either when transgenic CD8+ T cells are transferred or when primed CD8+ T cells are boosted to elicit a large secondary immune response.

A closer look at the results in **Figure 3E-G** (for the response with OT-I) versus **Figure 4B-D** (for the endogenous response) proved contrasting frequencies of SIINFEKL-specific IFN- γ -secreting cells. In those animals receiving OT-I cells, at day 14 post-immunisation, the spleen harboured 15 times more SIINFEKL-specific IFN- γ -secreting cells, with an average of 107,000 cells per mouse in **Figure 3G** after receiving 10,000 OT-I cells, while only 7100 cells were identified in those mice not receiving augmentative OT-I cells (**Figure 4D**). Our system works under the same premise as that shown by Kearney et al., 1994 (PMID: 7889419), where by adoptively transferring a number of transgenic cells, small populations can be more easily visualised by flow cytometry but that the ratio of transgenic cells is not too large to hide what would be normally seen.

We fully concur with R1 that the proliferation experiment (**Figure 2D-E**) could have been performed using congenic mouse strains in order to track only those initial CFSE+ cells, and we will certainly consider this suggestion for future work. Nonetheless, in our proliferation experiments (**Figure 3 and 4**) we provide robust evidence that the difference in frequencies (>10-fold) in those that received OT-I cells and those that did not can be attributed to proliferating OT-I cells. Also, the trend of frequencies shown in **Figure 2** are consistent with the outcomes with the rest of the initial figures: that a sporozoite surface antigen induces a higher CD8+ T cell response than EEF vacuolar membrane antigen.

Whilst we earnestly agree with R1 that it will be of interest to fundamentally explore further the fine specificities of the CD8+ T cell response induced by the transferred transgenic versus endogenous CD8+ T cells using the congenic system, we believe that this is beyond the scope of the current work. The paucity of such information does not modify the conclusions of our work concerning the relationships amongst the timing of antigen-specific expression, the induction of parasite-evoked antigen-specific CD8+ T cell responses and the generation of sterilising immunity from vaccine-induced antigen-specific CD8+ T cell responses.

As a compromise and as suggested by the editorial board, we have toned down some of our statements with regards to proliferation. We have removed the words 'early proliferative capacity' in the title preceding the paragraph describing the results of **Figure 2** (line 155-157). We also have added a few sentences on the importance of the congenic system in understanding the fine specificities (proliferative capacity, phenotypic and functional properties) of both the transferred and endogenous CD8+ T cells in the resulting immune response in the discussion on lines 361-373 and 424-425.

For **Figure 6**, as previously discussed, we did not vaccinate mice in the absence of OT-I cells (endogenous response). However, it is important to highlight that previous work with *Py*CSP and *Pb*CSP-based adenovirus vaccines generated endogenous CD8+ T cell responses that yielded only up to 40% sterile immunity, even with up to 97% reduction in parasite load in the liver after challenge with viable sporozoites (Rodrigues et al, 1997, PMID: 9013969; Rodrigues et al., 1998, PMID: 9795385). It is now well-established that a large reduction in parasite liver load does not imply sterilizing immunity, and this has to be tested in parallel experiments (as shown in **Figure 6**). Thus, the transfer of OT-I ensured the induction of optimal levels of vaccine-induced IFN- γ -producing [*effector*] CD8+ T cells in a single vaccination, with the ensuing frequencies comparable to those obtained by prime-boost vaccinations (from the same laboratories mentioned above) with recombinant adenovirus followed by vaccinia virus or alternative strategies, such as peptide-loaded dendritic cells followed by recombinant *Listeria*). We have added this information in the discussion to clarify this issue (line 380-387). Previous work by colleagues and ourselves have demonstrated that in order to achieve sterile protection in both the *Py*- and *Pb*-Balb/c model, vaccine-induced CSP-specific CD8+ T cells must exceed a threshold of >1% of all CD8+ T cells in peripheral blood (Bruña-Romero et al., 2001, PMID: 11553779; González-Aseguinolaza et al., 2003, PMID: 14557672; Schmidt et al., 2011, PMID: 21460205; Doll et al., PMID: 27084099). Yet, B10 backgrounds (including the C57BL/6 we used in our experiments) further increases the threshold required for sterile protection through a CD8+ T cell-extrinsic mechanism. In our current study, the mean frequencies of antigen-specific CD8+ T cells generated following adenovirus vaccination (with OT-I transfer) was 7.5% (**Figure 6C**), which translated to 80% sterile protection (combined data from CS^{SIINFEKL} and UIS4^{SIINFEKL} groups). Accordingly, the lack of data whether the CD8+ T cell response comes from the transferred or endogenous CD8+ T cells does not change how the single vaccination system in the presence of OT-I cells was utilised to evoke high levels of vaccine-induced CD8+ T cells required for protection.

2. R1 also requested to provide further support for the peptide-specificity in our intracellular cytokine staining assays.

We have reasoned the use of intracellular cytokine staining assays in quantifying IFN- γ production from endogenous CD8+ T cells in Point 1. We used a protocol that we have previously published to quantify CSP-specific responses in the BALB/c model (Gibbins et al., 2020; PMID: 32719159) and TRAP-specific (and others) responses in the C57BL/6 model (Hafalla et al., 2013; PMID: 23675294; Müller et al., 2017; PMID: 28380250). As requested by R1 previously, we also had included gating strategies and graphs displaying IFN- γ production from splenocytes and liver infiltrating lymphocytes not stimulated with peptides in **Appendix Figure S2** and **S3**, to show the background response. We rest assured that the peptide specificity is supported by robust evidence.

3. R1 would like to see gating strategies performed with and without the Live/Dead marker

We consistently perform experiments with and without using a Live/Dead marker. In the gating strategy examples provided below, we assayed splenocytes from mice that had been immunised with irradiated *P. berghei* WT sporozoites and spleen harvested 33 days post immunisation. We restimulated with *P. berghei* peptides (TRAP (SALLNVNDL) and S20 (VNYSFLYLF) from Hafalla et al. 2013 (PMID: 23675294) and NCY (NCYDFNNI) from Lau et al., 2014 (PMID: 24854165) or the irrelevant ovalbumin peptide SIINFEKL (neg OVA) or no peptide restimulation (neg med) for 5 hours and performed intracellular cytokine staining for IFN- γ and TNF, also staining extracellularly for CD3, CD8 and CD11a.

Using the same flow cytometry panel throughout, we incorporated a Live/Dead stain: i) as a separate first stain; ii) as part of the extracellular stain; iii) as a separate stain after the extracellular staining, as well as running the staining procedure without any Live/Dead staining. The plots in Figure A-C show the gating strategies (after singlet determination) and the proportion of cells ascertained at each stage with the different staining methods. The top row of plots shows Live/Dead (if gated for), Lymphocytes and CD8+ T cells (CD3+, CD8+) then the bottom row of plots shows CD11a+ IFN- γ + CD8+ T cells from cells restimulated with peptides: TRAP (left), SIINFEKL (neg OVA, middle) or no peptide restimulation (neg med, right).

D

E

Figure A shows an example gating strategy from where there is no Live/Dead dye included in the staining panel (and thus is not gated for). Figure B shows an example gating strategy from a staining panel where Live/Dead is incorporated into the extracellular staining and viable cells are gated before lymphocytes. Figure C shows an example gating strategy from the same staining panel as in Figure B, where Live/Dead is incorporated into the extracellular staining, but where the Live/Dead dye is not analysed and thus not gated for.

Based on the examples given above, that irrespective of whether there is a Live/Dead dye used and gated for (Figure D), a Live/Dead dye is employed but not used in the gating strategy (Figure E), or no Live/Dead marker is used (Figure E, first set of maroon bars), the proportions of antigen-specific CD8+ T cells derived are comparable.

4. R1 asked for clarification on ‘how many mice were used in each experiment’

We have now clarified this in the revision and added the information in the the figure legends and appendices. We have also included statements in the Materials and Methods regarding sample size choice, blinding, randomisation and the use of institutional care protocols.

5. R1 asked to rewrite their material and methods ... the part of lymphocytes isolation from livers (no Percoll gradient was apparently used to isolate lymphocytes or Did they used only liver cells as stated’

This is an important point and we apologise for this omission. We isolated intrahepatic or liver-infiltrating lymphocytes using the methods by Goossens et al., 1990 (PMID: 2202764) that utilises Percoll. We have expanded the portion of the materials and methods based on the above (lines 497-500).

6. R2 asked to discuss possible scenarios with ‘much less IFN+ CD11a+ specific CD8+T are induced’ by vaccination.

As we have alluded in Point 1, a much higher threshold for antigen-specific IFN- γ -producing CD8+ T cells is needed for sterilising protection in the B10 backgrounds (including the C57BL/6 we used in our experiments) as compared to BALB/c background. We have shown in the current study that sterile immunity is achievable in the C57BL/6 model. Clearly, further experiments are warranted to define this threshold, but this is beyond the scope of the current study. It is plausible that with fewer effector cells, sterile protection will not be achieved. A translational challenge for T cell-based malaria vaccines will be both the generation and maintenance of optimal levels of effector CD8+ T cells required for protection and we hope that our study inspires interdisciplinary research into T-cell vaccines.

7. R2 requested for ‘quantification and statistics on the levels of CD11a , CD62L , CD49d and the proliferation of CFSE-labelled cells’

We have now added the quantification and statistics for the levels of K^b/SIINFEKL+ CD8+ T cells that individually co-stain with K^b-SIINFEKL and CFSE^{lo}, CD11a^{hi}, CD62L^{lo} or CD49d^{hi}. These are now shown in **Figure EV2**.

8. R2 suggested that ‘title of the figures could be more consistent in form, sometimes being descriptive of the experiment’

We have now changed some titles of the figure legends and titles in the results section of the text to reflect outcomes, when possible. Please see revised titles for legends to **Figure 2** and **4** (line 776-777 and 814-815) and results subtitle when discussing **Figure 4** on line 233-234.

9. R2 suggested some ‘In 50 needs a comma between of and and in

We thank R2 for this comment. Please see alteration on line 54.

10. R2 suggested that 'in 216 "visualise" would probably be better replaced by identify or detect'

We thank the reviewer for this suggestion. Please see alteration on line 248.

11. R2 spotted that 'in 52 25 vs In 271 35 years of livers age vaccines, although both are true, would probably be better to go for 35 in both instances'

We thank R2 for spotting this inconsistency. This has been altered to "35 years" on line 56 and 310.

12. R3 spotted 'a minor problem with the sentence in the table legend "b brackets indicate the only a small proportion of mice...." (Comment 19)'

We thank R3 for spotting this error. Please see the alteration on line 939.

13. R3 requested to comment on 'It is even clearer from the new Figure S1E that CSP-SIINFEKL display lower EEF numbers ... state if not significant ...'

We have performed and added the statistics for **Figure EV1E** using one-way ANOVA and Tukey's multiple comparison test. The numbers of EEFs in the different groups are not statistically different from each other except at 72 hours, where the numbers of CSP^{SIINFEKL} EEFs are significantly lower than WT. We have mentioned the use of the test in the figure legend of **Figure EV1** on line 892-893. Nonetheless, we do not think that this shows that there is an obvious defect in the parasite growth and development as numbers remain non-significantly different at 24 and 48 hours, after which point, most of the EEFs would have been released as merozoites during normal liver maturation *in vivo*.

14. R3 suggested 'Would be good to now include similar confirmation of EEF numbers (based on microscopy) as a complement to qPCR data presented in Figure 6D if available, as requested initially, but not a prerequisite to publication'. (Comment 20).

Unfortunately, we do not have additional confirmation of EEF numbers by microscopy. The qPCR data we presented in **Figure 6D** is based on a standard methodology widely utilised by different groups in the field of malaria pre-erythrocytic stages. Whilst microscopy is a standard method to count EEFs in culture systems, it remains challenging to perform *in vivo* and is less suitable than qPCR for accurate quantification of liver parasite loads in mice.

24th Dec 2020

Dear Dr. Hafalla,

Thank you for the submission of your revised manuscript to EMBO Molecular Medicine. I am pleased to inform you that we will be able to accept your manuscript pending the following final amendments:

1) With approaching holidays and the end of the year we encountered high number of submissions, so that our data editors were not able to process all received manuscripts before the holiday season. Therefore, we will send you the document with data editor's suggestions after the holidays and as soon as our data editors process your manuscript. Please do not submit your revised manuscript before we send you the file with data editor's suggestions. Thank you for your understanding.

2) Please address all the points raised by the referee #1. No additional experiments are required, however, as suggested by the referee #1 further toning down the conclusions and acknowledging the limitations of the study are necessary.

***** Reviewer's comments *****

Referee #1 (Remarks for Author):

The third time I review the manuscript from Mueller et al., and I still think that this study has potential and would constitute an important study in malaria vaccine files. Nevertheless, the authors repeatedly refused to use essential immunology tools to improve the quality of their work. THIS STUDY AND ITS CONCLUSIONS HAVE TO BE TONE DOWN TO BE ACCEPTED, AND THE AUTHORS SHOULD ACKNOWLEDGE THE LIMITATIONS. Here are the comments to the authors' responses to the second round questions:

1.1

This reviewer is well aware of the published work from leading laboratories in the field. Nevertheless, I still do not understand how the authors refuse to address a question as simple as to use CD45.1 congenic mice to track the CD8 OTI responses. The cited leading laboratories also used this system in their studies, and it is a standard procedure in immunology.

Moreover, argument-to-authority is contemptuous and disrespectful.

1.2

Still, the authors are speculating about the OTI cells in figure 3E-G and Fig 4D. They cannot know if those are OT-I or not. Please tone down the conclusions and acknowledge the limitations of the study as is.

1.3 Again, authors are speculating about the responses they observe. Thus I would suggest that they tone down their conclusions if they agree that proliferation experiments could have been appropriately performed. A better solution would be to perform those experiments

1.4 Concerning

figure 6, the Authors said (380-385) that "transfer of OTI ensured the induction of optimal levels of vaccine-induced effector CD8+T cells. Control is still missing here (vaccinated mice without OTI transfer). How can they know that protection is due to the SIINFKL- transgenic parasites recognized by the endogenous. If a CD8 T cells transfer is required to warrant protection, the conclusion would be different. I understand that the epitope recognized by OTI transfer cells but which one? The one generated by vaccination or both?

2.Thanks, I appreciate this. Furthermore, I am convinced now by the gating.

3.I appreciate

the effort demonstrating that not using the Live/Dead marker equivalent to use the Live/Dead marker. I still think that this should be performed more exhaustively. Besides, such marker usage is a gold standard on cytometry (used by leading or not laboratories worldwide). The authors would not have to convince this reviewer of their choice if they had used it initially. The figure below is only one experiment; how can they be sure that in each experiment performed, they have the same healthy cells?

The authors performed the requested changes.

We express thanks to the EMBO Editorial Board for communicating the outcome of their internal review and another round of comments from Reviewer 1. We heeded the advice of EMBO Molecular Medicine that no additional experiments are required. We thank Reviewer 1 for their arduous and attentive review of our manuscript; we appreciate their acceptance of our gating strategy.

Whilst we appreciate the reviewer's recommendations for future experimentation, we maintain that the paucity of the information accounting for the difference between endogenous and transgenic responses does not alter the conclusions of our work regarding the associations amongst the staging of antigen-specific expression, the generation of parasite-evoked antigen-specific CD8+ T cell responses and the induction of complete protection due to vaccine-induced antigen-specific CD8+ T cell responses., we believed that we have toned down our statements in the last round of revisions. Nonetheless, in this submission, we have done the following:

- Line 194, we have removed the quantification of difference between the primary responses to CSP^{SIINFEKL} versus UIS4^{SIINFEKL}. This change was done as not to denote the "6x difference" as referring to the proliferation experiment.
- Line 384: we have added the word "versus" to emphasise Reviewer 1's point of fundamentally comparing responses in the presence of absence of OT-I cells.
- Line 386: we have added the word "limitations" as a recognition that the issue is a limitation of the current work.

We agree with Reviewer 1 that transferring OT-I cells prior to vaccinating mice with recombinant Adenovirus creates an uncertainty as to the requirement for these cells in order to achieve complete protection. In our previous revision, we elaborated on the use of prime-boost approaches to attain the numbers of CD8+ T cells required for protection. In addition to toning down conclusions in the previous revision, we now have done the following:

- Line 396, we have replaced the word 'optimal' with 'high'.
- Line 404-408, we have added a statement describing previous studies with CSP-based adenoviral vaccines and endogenous responses. The statement reads as follows: "Previous work in mouse models and CSP-based adenovirus vaccines showed the generation of endogenous responses that yielded only up to 40% sterile immunity, despite two orders of magnitude in the reduction in parasite load in the liver after challenge with viable sporozoites (Rodrigues et al, 1997, PMID: 9013969; Rodrigues et al., 1998, PMID: 9795385)". This further emphasises the contrast between endogenous responses following a single vaccination system and that of prime-boost approaches. The latter generates high levels of CD8+ T cell responses required for protection. In our experiments, OT-I cells were transferred in order to achieve high levels of responses.
- Line 415-419 – We have added that our work provides proof-of-concept data, and it was not our intention to convey a message that some sort of cell transfer must be performed for malaria vaccines to work. The statement reads: "Our work provides proof-of-concept for vaccines targeting all malaria pre-erythrocytic stages. The translational challenge will now be to design vaccine formulations, which evoke and maintain high levels of antigen-specific human CD8+ T cells either given as single dose or as part of a prime-boost approach".

In addition to the above revisions, we have updated the Graphical Abstract to make it less busy. We have changed 'Synopsis Summary' to be in passive voice. We also have added the missing "This Paper Explained". We have updated the Materials and Methods to include relevant sentences from the Author's Checklist, as well as included the antibody dilutions used in lines 545-547, 556-559 and 575-591. We have added the statement "See Appendix Table S4 for exact p-values" after each statistical test that is mentioned in the figure legends and included a table of exact p values in the Appendix, as well the suggestions for the figure legends from the Data Editor.

5th Feb 2021

Dear Dr. Hafalla,

We are pleased to inform you that your manuscript is accepted for publication.

Corresponding Author Name: Julius Clemence R Hafalla

Manuscript Number: EMM-2020-13390